# Expression of AXL receptor tyrosine kinase relates to monocyte dysfunction and severity of cirrhosis

Robert Brenig[1,2] , Oltin T Pop[2,3], Evangelos Triantafyllou[3,4] , Anne Geng[1], Arjuna Singanayagam[3,4] , Christian Perez-Shibayama[2,5], Lenka Besse[6], Jovana Cupovic[5], Patrizia Künzler[2] , Tuyana Boldanova[1], Stephan Brand[2], David Semela[2], François HT Duong[1], Christopher J Weston[7] , Burkhard Ludewig[5], Markus H Heim[1] , Julia Wendon[3], Charalambos G Antoniades[3,4], Christine Bernsmeier[1,2]

**Infectious complications in patients with cirrhosis frequently initiate episodes of decompensation and substantially contribute to the high mortality. Mechanisms of the underlying immuneparesis remain underexplored. TAM receptors (TYRO3/AXL/MERTK) are important inhibitors of innate immune responses. To understand the pathophysiology of immuneparesis in cirrhosis, we detailed TAM receptor expression in relation to monocyte function and disease severity prior to the onset of acute decompensation. TNF-$\alpha$/IL-6 responses to lipopolysaccharide were attenuated in monocytes from patients with cirrhosis (n = 96) compared with controls (n = 27) and decreased in parallel with disease severity. Concurrently, an AXL-expressing (AXL[+]) monocyte population expanded. AXL[+] cells (CD14[+]CD16[high]HLA-DR[high]) were characterised by attenuated TNF-$\alpha$/IL-6 responses and T cell activation but enhanced efferocytosis and preserved phagocytosis of *Escherichia coli*. Their expansion correlated with disease severity, complications, infection, and 1-yr mortality. AXL[+] monocytes were generated in response to microbial products and efferocytosis in vitro. AXL kinase inhibition and down-regulation reversed attenuated monocyte inflammatory responses in cirrhosis ex vivo. AXL may thus serve as prognostic marker and deserves evaluation as immunotherapeutic target in cirrhosis.**

## Introduction

Patients with cirrhosis are at increased risk of infection and consequent acute decompensation (AD) with substantially elevated morbidity and mortality (1). Compared with the overall rate of infections in hospitalised patients (5–7%), bacterial infections occur significantly more frequently in patients with cirrhosis (32–34%) (2, 3).

Similarly, infections account for more than 50% of hospitalisations in cirrhotic patients, are the main precipitant for AD without and with organ failure (acute-on-chronic liver failure [ACLF]) (4, 5), and implicate a high mortality (2, 6). Infection susceptibility in cirrhosis has been attributed to a state of immuneparesis, defined by inadequate immune responses to microbial challenge (7, 8, 9).

The pathophysiology of immuneparesis in cirrhosis is highly complex and remains incompletely understood, involving diverse defects in immune cell function, including monocytes, and soluble factors in multiple compartments (7). Circulating monocytes from patients with AD and ACLF compared with stable cirrhosis demonstrated reduced expression of HLA-DR and attenuated production of TNF-$\alpha$/IL-6 in response to lipopolysaccharide (LPS), which has previously been linked to adverse outcome (10, 11, 12). Moreover, the role of bacterial translocation in the pathogenesis of immune dysfunction and infection susceptibility has been highlighted (13, 14).

TAM receptors (TYRO3, AXL, and MERTK) belong to the family of receptor tyrosine kinases. Among immune cells, they are expressed on monocytes, macrophages, dendritic cells, and glial cells, and additionally on epithelial cells of the reproductive system, the retina, and tumour cells (15). TAM receptors are important regulators of innate immune homeostasis, acting by inhibition of TLR signalling pathways through a signal transducer and activator of transcription 1 (STAT1)- and suppressors of cytokine signalling (SOCS1/3)-dependent mechanism (15, 16) and by promotion of phagocytic removal of apoptotic cells (efferocytosis) (16). Their activation succeeds ligand binding (growth arrest–specific gene-6 [GAS6], PROTEIN S) and interaction with phosphatidylserine on apoptotic cells (15, 16, 17). In murine dendritic cells, activation required interaction with the type I interferon receptor (IFNAR) (16).

We recently identified the expansion of MERTK-expressing monocytes and macrophages in diverse compartments in patients with ACLF that dampened innate immune responses to microbial challenge and

[1]Department of Biomedicine, University of Basel and University Centre for Gastrointestinal and Liver Diseases, Basel, Switzerland   [2]Medical Research Centre and Division of Gastroenterology and Hepatology, Cantonal Hospital St. Gallen, St. Gallen, Switzerland   [3]Institute of Liver Studies, King's College Hospital, King's College London, London, UK   [4]Hepatology Department, St. Mary's Hospital, Imperial College London, London, UK   [5]Institute of Immunobiology, Medical Research Centre, Cantonal Hospital St. Gallen, St. Gallen, Switzerland   [6]Laboratory of Experimental Oncology, Department of Oncology and Haematology, Cantonal Hospital St. Gallen, St. Gallen, Switzerland   [7]Centre for Liver Research and National Institute for Health Research, Biomedical Research Unit, University of Birmingham, Birmingham, UK

Correspondence: c.bernsmeier@unibas.ch

conferred disease severity and adverse outcomes (18). The expansion of MERTK-expressing monocytes and macrophages was moreover detected in acute liver failure (18) and characterised by both suppressed immune responses and enhanced efferocytic capacities (19). Another immune-suppressive population, expanded in the circulation of patients with ACLF, was monocytic myeloid–derived suppressor cells (M-MDSC) that suppressed T cell activation, innate immune responses, and pathogen uptake (20).

It is not clear when and/or under which circumstances immuneparesis and monocyte dysfunction occurs with an associated susceptibility to infection during the clinical course of cirrhosis and portal hypertension, before the onset of AD. The main emphasis of this study is to detail the expression of TAM receptors on monocytes in relation to monocyte function and disease severity of patients with cirrhosis in the absence of AD using patients with AD, chronic liver disease (CLD) without cirrhosis and healthy controls (HCs) as comparators. We hereby seek to better understand the pathophysiology of immuneparesis development in patients with cirrhosis prior to AD and identify candidates for biomarkers and future immunotherapeutic targets that may preserve innate immune responses.

## Results

### Patient characteristics

Patients with cirrhosis were distinguished between Child-Pugh A, B, and C and compared with AD, CLD without cirrhosis, and HC. The cohort was characterised by disease severity scores, aetiologies, and diverse clinical parameters (Tables S1 and S2). In patients with cirrhosis without AD, 1-yr mortality rate was 4.5% and rising with Child-Pugh stage: A (0%), B (5.7%), and C (11.8%). 1-yr mortality rate for those with AD was 75%, with two of eight patients dying within 28 d of enrolment. N = 9 patients deceased from cirrhosis-related complications during follow-up of 1 yr (secondary infections [n = 4], ACLF with multiorgan failure [n = 3], hepatocellular carcinoma [HCC] [n = 1], hypovolemic shock due to variceal bleeding [n = 1]), and the cause of one death was unknown. Current infections at hospital admission were seen in 62.5% of patients with AD. Within 4 wk following inclusion into the study, 5.2% of patients (Child B: 5.7%, Child C: 5.9%, and AD 25%) developed infectious complications, adapted from the definition by Bajaj et al (9). Episodes of AD developed in 10.2% of cirrhotic patients (Child B: 17.1% and Child C: 17.6%) within 4 mo following inclusion (Table S1).

### Innate immune responses are impaired in patients with cirrhosis and parallel the expansion of an AXL-expressing circulating monocyte population

In patients with AD/ACLF, we recently described impaired inflammatory cytokine production of circulating monocytes to microbial challenges (18, 20). Attenuated responses were also seen in stable cirrhotic patients (18, 20). It, however, remained unknown when and to what extent circulating monocytes develop immune dysfunction over the time course of disease progression. We measured ex vivo inflammatory cytokine production upon LPS treatment of circulating monocytes from patients with cirrhosis at different stages of disease. TNF-α and IL-6 production was reduced in cirrhosis compared with HC and incrementally decreased from Child A to C, and AD but remained preserved in patients with CLD without cirrhosis (Fig 1A and B).

In parallel with increased disease severity and the decline of inflammatory cytokine production in response to LPS, we demonstrated the expansion of an AXL-expressing monocyte population ex vivo in the circulation of patients with cirrhosis (Figs 1C and S1A). The occurrence of AXL-expressing monocytes was independent of the underlying aetiology and other potential confounders (inpatient treatment, current infection, antimicrobial treatment, immunosuppressive therapy, and non-metastatic malignancies; Fig S1B and D). Within monocyte subsets, the expression of AXL was highest in but not restricted to the intermediate subset (cluster of differentiation [CD]14$^{++}$CD16$^+$) (Fig S2A). AXL expression on monocytes of patients with CLD without cirrhosis was low; a similar pattern was also seen in AD (Fig 1C). Other immune cells such as lymphocytes and granulocytes barely expressed AXL (Fig S2B). Longitudinal follow-up data showed an increase in AXL expression after re-compensation of AD episodes and a change in AXL expression paralleling the evolution of disease severity after 1 yr (Fig S1E and F). Recently, we described a MERTK-expressing monocyte population that was expanded in the circulation of patients with AD/ACLF (18), which was again confirmed in this cohort (Fig 1D). In CLD with and without compensated cirrhosis, however, MERTK and TYRO3 expressions were sparse (Figs 1D and E, and S1A). Circulatory plasma levels of the AXL ligand GAS6 were significantly elevated in cirrhosis compared with HC, independent of the aetiology. GAS6 increased from Child A to C and correlated with AXL-expressing monocytes (Figs 1F and S1C).

### Circulating AXL-expressing monocytes in patients with advanced cirrhosis indicate diseases severity, complications, and poor outcome

We next assessed the expansion of AXL-expressing monocytes in relation to clinical parameters, disease severity scores, indicators of complications, and outcome. The proportion of AXL-expressing monocytes strongly correlated with Child-Pugh and model for end-stage liver disease (MELD) scores and the classification of cirrhosis established by D'Amico et al (21) (Fig 2A). AXL-expressing monocytes also correlated with soluble AXL (sAXL) plasma levels. sAXL was significantly elevated in cirrhosis compared with controls, independent of the underlying aetiology, correlated with Child-Pugh and MELD, and predicted the onset of AD episodes within 4 mo following study inclusion (Fig S3A–E).

AXL on monocytes predicted 1-yr mortality with a sensitivity of 80% and specificity of 79.2% for the criterion median fluorescence intensity (MFI) > 440. AXL moreover predicted the onset of AD within 4 mo following inclusion, and the development of infection over the next 4 wk. AXL was also associated with C-reactive protein (CRP) (Figs 2B and S4A, and B). Furthermore, AXL-expressing monocytes were associated with manifestations of portal hypertension (ascites, hepatic venous pressure gradient, varices, hepatic encephalopathy,

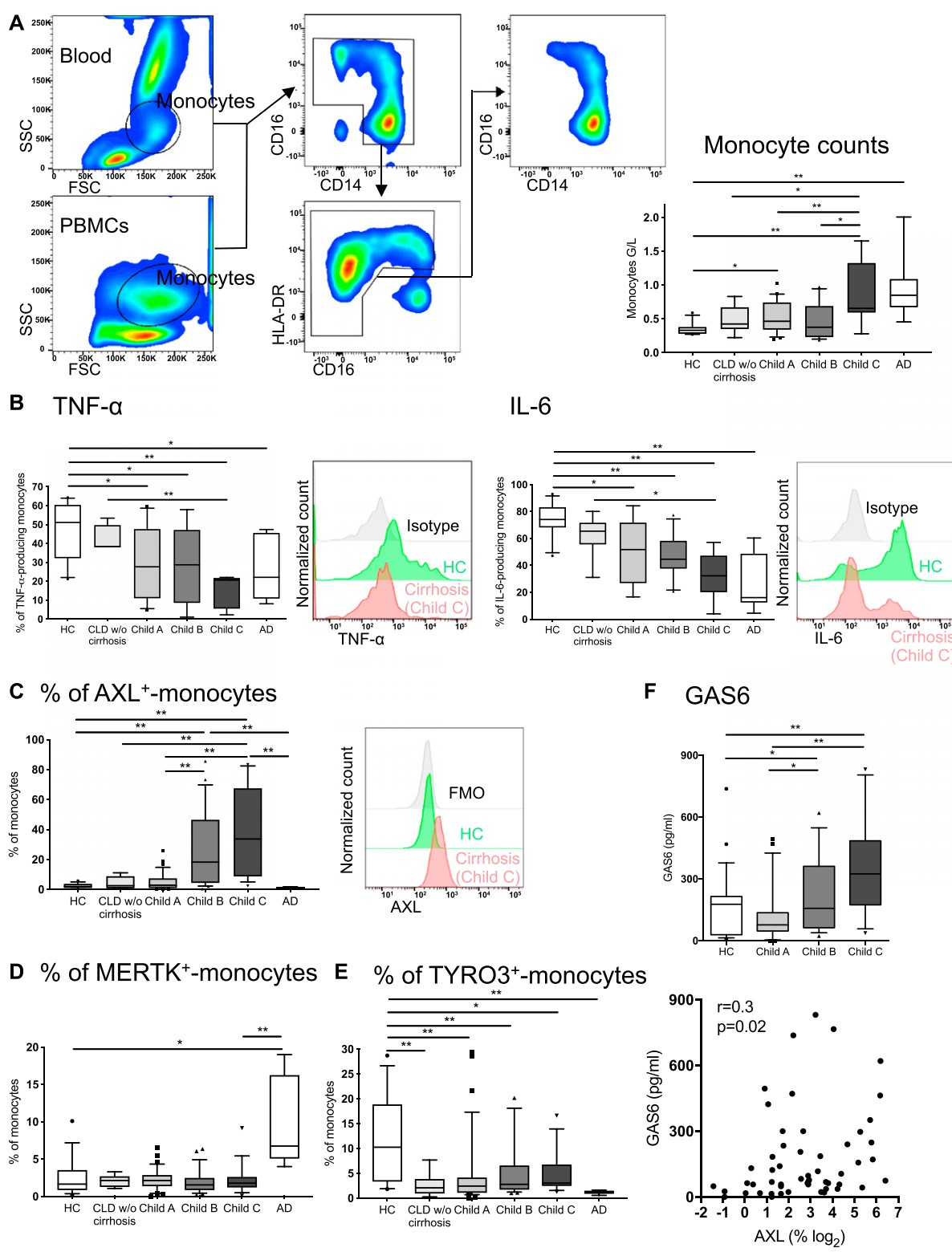

**Figure 1. TAM (TYRO3, AXL, and MERTK) receptor expression and functional characterisation of circulating monocytes in cirrhosis.**
**(A)** FACS gating strategy used to identify circulating monocytes in whole blood or PBMCs (left panel). Side scatter (SSC), forward scatter (FSC). Monocyte counts (differential leucocyte count, right panel). **(B)** TNF-α– and IL-6–producing monocytes (%) in response to LPS ex vivo at different stages of cirrhosis and representative FACS histograms (HC, Child C, and isotype). **(C, D, E)** TAM receptor expression on circulating monocytes (%) at different stages of cirrhosis and representative FACS histograms for AXL expression (HC, Child C, and fluorescence minus one). **(F)** GAS6 levels (pg/ml) in HC and cirrhosis (upper panel) and in correlation with AXL expression (% of monocytes, lower panel). HC n = 27, CLD without (w/o) cirrhosis n = 8, Child A n = 36, Child B n = 28, Child C n = 17, and acute decompensation (AD) of cirrhosis n = 8. Data are presented as box plots showing median with 10–90 percentile. *P < 0.05/**P < 0.01 (Mann–Whitney tests, Spearman correlation coefficient).

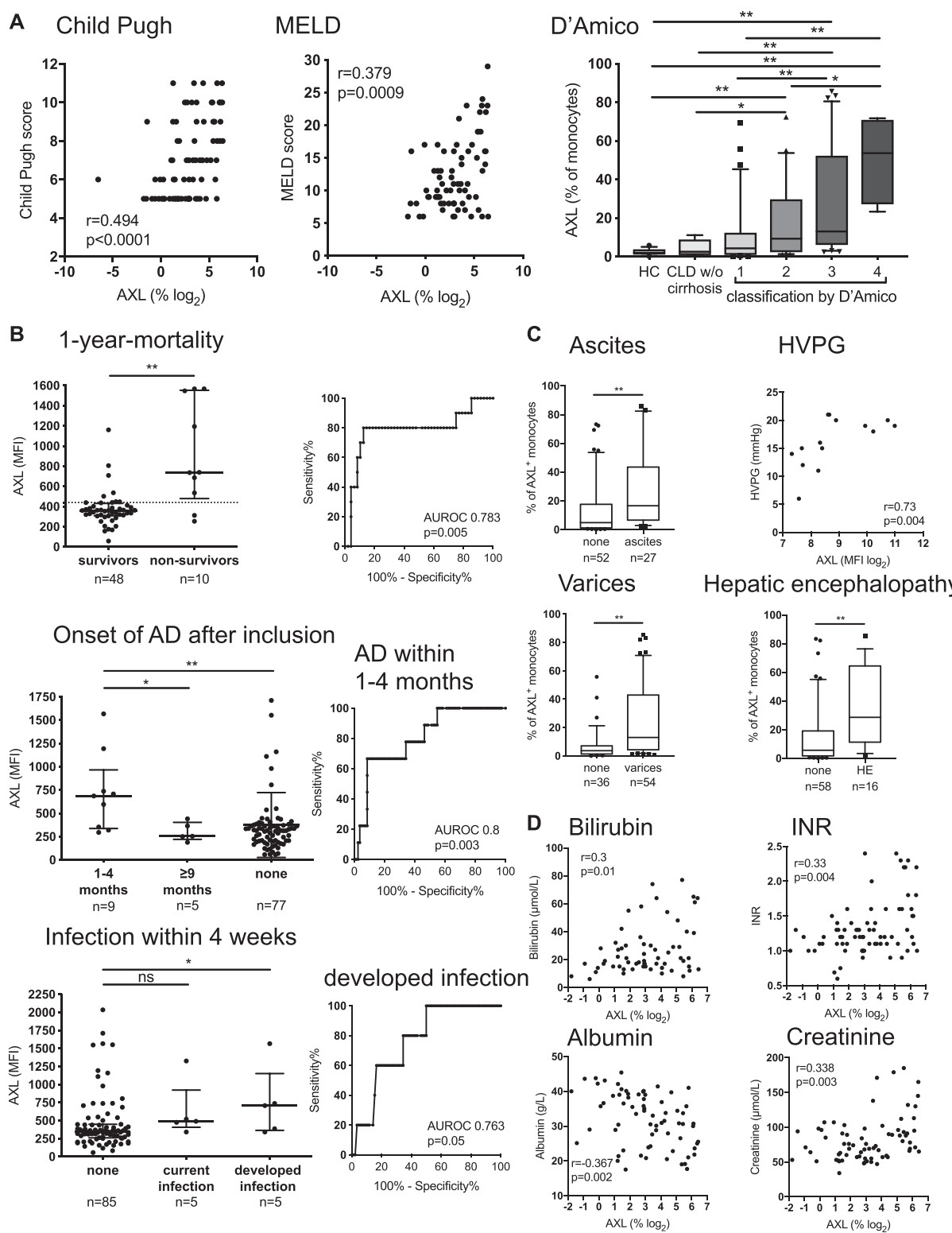

**Figure 2. The AXL-expressing monocyte population in patients with cirrhosis in relation to disease severity, complications and prognosis.**
**(A)** Correlations of AXL-expressing monocytes (%) with Child-Pugh (n = 78) and MELD (n = 73) scores and the classification by D'Amico et al ([21]). HC, CLD without (w/o) cirrhosis. Box plots showing median/10–90 percentile. **(B)** AXL expression predicted 1-yr mortality (criterion MFI > 440, sensitivity 80%, specificity 79.2%), development of further episodes of AD of cirrhosis within 4 mo (criterion MFI > 362, sensitivity 66.7%, specificity 67.1%), and development of infection over 4 wk (criterion MFI > 389, sensitivity 60%, specificity 65.5%). Median/interquartile range (IQR). **(C, D)** AXL-expressing monocytes in relation to portal hypertension (C: ascites, hepatic venous pressure [HVPG, n = 14], varices, hepatic encephalopathy; D: bilirubin, n = 72; INR, n = 74; albumin, n = 72; and creatinine, n = 75). Median/10–90 percentile. *P < 0.05/**P < 0.01 (Mann–Whitney tests, Spearman correlation coefficient).

and renal dysfunction) and correlated with individual parameters of liver function (bilirubin, international normalised ratio [INR], albumin) (Fig 2C and D). High AXL expression on monocytes may albeit small numbers also predict the need for transplantation, transplantation-free 1-yr survival, and development of HCC within 1 yr (Fig S4C–E).

## Phenotype of circulating monocytes in patients with cirrhosis and the AXL-expressing monocyte population

Monocytes from patients with CLD without cirrhosis did not differ phenotypically from HC. Monocytes from patients with cirrhosis, however, showed an HLA-DR$_{low}$ phenotype with decreased expression of Fcγ- and homing receptors (CD32$_{low}$CX3CR1$_{low}$CCR7$_{low}$). HLA-DR expression significantly decreased from Child A to C (Fig S5A).

Within this entire population, the expanded subset of AXL-expressing monocytes (AXL$^+$) (Fig 3A) were CD14$^+$CD16$^{high}$HLA-DR$^{high}$ indicating a mature monocyte subpopulation with augmented expression of Fcγ-receptor CD32, TLR4, and homing/chemokine receptors (CCR5, CCR7, and CX3CR1) (Fig 3B). There was no difference in viability between AXL$^+$- and AXL$^-$-monocytes (Fig 3C).

Importantly, the CD14$^+$HLA-DR$^+$AXL$^+$ immune cell subset detailed here has to be distinguished from the recently identified immunosuppressive M-MDSCs in patients with cirrhosis and ACLF (20), which we observed expanding from Child A to C in our cohort. M-MDSCs were CD14$^+$CD15$^-$CD11b$^+$HLA-DR$_{low/neg}$, as previously defined (22), and expressed lower levels of AXL in comparison with CD14$^+$HLA-DR$^+$ monocytes (Fig S6A–C).

## AXL-expressing circulating monocytes contribute to impaired innate immune responses and suppression of T cell proliferation while retaining phagocytic capabilities for bacteria ex vivo

To investigate the effect of the expanded CD14$^+$HLA-DR$^+$AXL$^+$ subset on the impaired inflammatory cytokine responses observed in monocytes from patients with cirrhosis we assessed the functional properties of AXL$^+$ monocytes ex vivo.

Detailed analyses of the distinct subsets revealed that TNF-α/IL-6 production in response to LPS was decreased in both CD14$^+$HLA-DR$^+$AXL$^+$ monocytes and M-MDSCs when compared with CD14$^+$HLA-DR$^+$AXL$^-$ monocytes from patients with cirrhosis and HC. In detail, TNF-α production decreased from 59.2% (13.8) to 40% (34.6) of monocytes in CD14$^+$HLA-DR$^+$AXL$^-$ from HC versus CD14$^+$HLA-DR$^+$AXL$^+$ from patients (MFI: 2665 [1668] versus 1117 [2812]) and IL-6 from 73.7% (37.3) to 40% (55) (MFI: 1909 [1702] versus 408 [298]; median [interquartile range, IQR]) Figs 4A and S7A, and B). The CD14$^+$HLA-DR$^+$AXL$^-$ population represented the majority of monocytes in HC, indicating it may be regarded "functionally intact" but was sequentially lost in the circulation of patients with progression of cirrhosis (Fig S6B). In line with previous data detailing TAM receptor signalling pathways (16), we observed higher mRNA levels of SOCS1/SOCS3 in monocytes of Child B/C patients, compared with HC (Fig 4B). Our data thus reveals the determination of functional roles of monocyte subsets in a pathophysiological context such as cirrhosis.

We further revealed that AXL expression on monocytes in cirrhosis was associated with inhibition of T cell proliferation, when tested in an allogeneic mixed lymphocyte reaction (Fig 4C).

Ex vivo phagocytic capacity of *Escherichia coli* (*E. coli*) bioparticles and live GFP-containing *E. coli* by circulating monocytes did not differ between cirrhotic patients and HC. AXL$^+$ monocytes showed preserved phagocytosis capacities for *E. coli* bioparticles and live *E. coli* bacteria, whereas M-MDSCs revealed reduced phagocytosis, when compared with CD14$^+$HLA-DR$^+$AXL$^-$ monocytes from patients with cirrhosis and HC (Figs 4D and S7C).

Considering these observations, the expanded CD14$^+$HLA-DR$^+$AXL$^+$ monocyte population in the circulation of patients with cirrhosis (notably, not existing in healthy subjects) remained functionally phagocytic, but prevented T cell proliferation and inflammation (low TNF-α/IL-6 production) in a presumably SOCS1/3-dependent manner, representing an immune-regulatory "homeostatic" monocyte population expanding during cirrhosis progression.

## AXL overexpression in THP-1 cells attenuates LPS-induced inflammatory cytokine production in vitro

As proof-of-concept for the observations developed above, in vitro, we overexpressed AXL in the monocytic THP-1 cell line using a retroviral system (Fig 5A). Following transduction, AXL mRNA expression (2.2 ± 0.3-fold; Fig S8A) and protein levels (88% THP-1-AXL$^+$ cells; Fig 5A and B) were increased. Phenotypic characterisation of the THP-1-AXL$^+$ model cell line is illustrated in Fig S8B. Consistent with the observations in patients with cirrhosis ex vivo, AXL-expressing THP-1 cells produced less TNF-α and IL-6 in response to LPS when compared with non-transduced THP-1 cells (Fig 5C).

## Pathogen-associated molecular patterns (PAMPs), cytokines, bacterial uptake, and efferocytosis induce AXL up-regulation on monocytes

Next, we sought to understand the mechanisms leading to the expansion of the described immune-regulatory AXL-expressing monocyte population. Pathophysiologically, cirrhosis progression involves development of portal hypertension and subsequent pathologic bacterial translocation facilitates microbial products accessing the systemic circulation (8, 13, 14, 23). Hence, we tested PAMPs and damage-associated molecular patterns (DAMPs) for their ability to modify monocyte differentiation and AXL expression in vitro. Stimulation with bacterial products such as TLR ligands (Pam3SK4, LPS, CpG, and poly (I:C)) significantly up-regulated AXL expression in vitro. Similarly, pro-inflammatory factors (IFN-α and TNF-α) induced AXL up-regulation. In contrast, the DAMP high-mobility group protein B1 (HMGB1), TGF-β, and AXL ligand GAS6 did not induce AXL expression. LPS-induced up-regulation of AXL was time dependent (Fig 6A), and those monocytes produced significantly less TNF-α/IL-6 upon LPS when compared with monocytes without prior LPS exposure (Fig S9A). Notably, monocytes incubated with 25% plasma of cirrhosis patients did not show biologically relevant changes in AXL expression (Fig 6B), suggesting that additional factors are required to generate this subset in vivo.

Phagocytosis is required for efficient clearance of pathogenic microorganisms and initiation of various immune responses. AXL$^+$

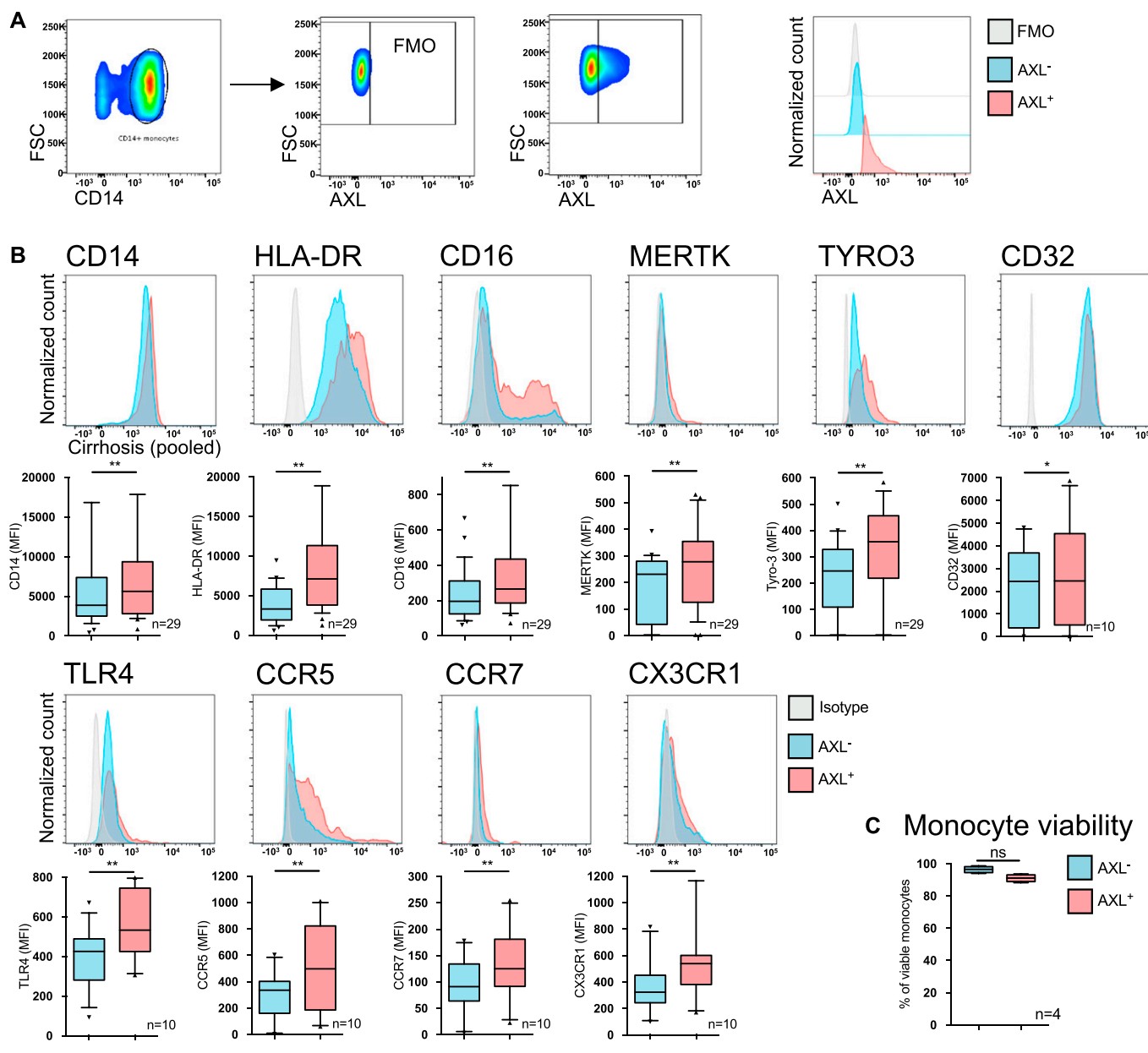

**Figure 3. Phenotypic characterisation of the AXL-expressing monocyte subset.**
**(A)** Gating strategy with representative FACS scatter plots and histograms for AXL expression used to distinguish AXL-expressing (AXL[+]) from AXL-negative (AXL[−]) monocytes. Side scatter (SSC), forward scatter (FSC), fluorescence minus one. **(B)** Immunophenotyping of AXL[+] and AXL[−] monocytes in cirrhosis. Glycoprotein CD14, MHC class II receptor HLA-DR, Fcγ-receptors (CD16 and CD32), TAM receptors (MERTK and TYRO3), chemokine receptors (CX3CR1, CCR5, and CCR7), and TLR4. Box plots showing median/10–90 percentile. **(C)** Viability (7-AAD[−]AnnexinV[−]-cells) of AXL[+]-/AXL[−]-monocytes. Median/10–90 percentile. *$P < 0.05$/**$P < 0.01$ (Wilcoxon tests).

monocytes exhibited preserved phagocytosis capacities (Figs 4D and S7C) and AXL expression significantly increased on monocytes after phagocytosis of *E. coli* and *Staphylococcus aureus* (*S. aureus*) bioparticles as well as of live GFP-*E. coli* bacteria (Fig 6C). Phagocytosis capacity of *E. coli* bioparticles positively correlated with the degree of AXL expression and concurrently negatively with TNF-α production (Fig S7D–E).

Within inflammatory milieus, as prevalent in different compartments of patients with cirrhosis (8), efferocytosis is required to maintain immune homeostasis (24, 25). We, therefore, co-cultured

healthy monocytes with neutrophils and HepG2 cells previously labelled with a cytoplasmic cell-tracker and then induced to apoptosis (19, 26). AXL expression on monocytes was up-regulated following co-culture with apoptotic cells. Monocytes that engulfed apoptotic cells (efferocytosing) were characterised by higher AXL expression, compared with monocytes that did not (resting). AXL[+] monocytes showed higher efferocytosis capacity than AXL[−] monocytes when co-cultured with apoptotic neutrophils (Fig 6D); following efferocytosis of neutrophils, AXL[+] monocytes produced less TNF-α/IL-6 upon LPS than AXL[−] monocytes (Fig S9B), supporting the

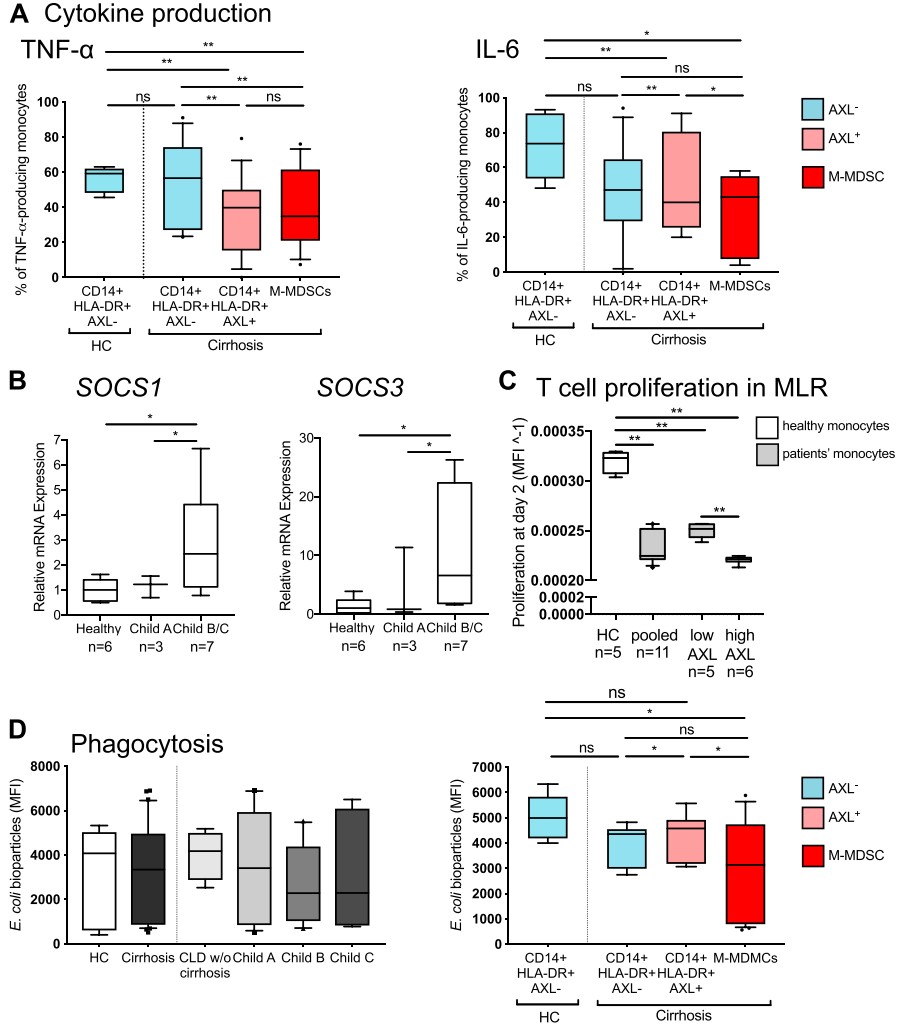

**Figure 4. Functional characterisation of AXL-expressing circulating monocytes ex vivo.**
**(A)** TNF-α and IL-6 production upon LPS treatment of CD14⁺HLA-DR⁺AXL⁺, CD14⁺HLA-DR⁺AXL⁻ monocytes and M-MDSCs from HC and patients with cirrhosis (% of monocytes). **(B)** SOCS1/3 mRNA-expression of monocytes from HC and cirrhosis. **(C)** T cell proliferation in co-culture with monocytes at day 2 in a mixed lymphocyte reaction (HC versus cirrhosis; AXL_low versus AXL^high). Data shown as MFI⁻¹ of carboxyfluorescein succinimidyl ester. **(D)** Phagocytosis of *E. coli* bioparticles of the entire monocyte population from different patient groups (HC, CLD without [w/o] cirrhosis, cirrhosis, left panel) and CD14⁺HLA-DR⁺AXL⁺, CD14⁺HLA-DR⁺AXL⁻ subsets, and M-MDSCs from HC and patients with cirrhosis. Box plots showing median/10–90 percentile. *P < 0.05/**P < 0.01 (Mann–Whitney, Wilcoxon tests).

accumulation of AXL⁺ immune-regulatory monocytes in an inflammatory environment.

## AXL inhibitor BGB324 and metformin restore innate immune responses of monocytes from patients with cirrhosis ex vivo

Given the distinct immune-regulatory functions of the AXL⁺ monocyte population in patients with cirrhosis and its association with disease severity and infection, we questioned whether inhibition or down-regulation of AXL would reverse the anti-inflammatory properties. BGB324 is a selective small molecule inhibitor of AXL previously tested in clinical studies (27). Metformin, a well-known antidiabetic drug, was previously described to down-regulate AXL expression (28) and to regulate the AXL signalling cascade in the context of cancer (29, 30). Metformin-induced down-regulation of AXL was confirmed in monocytes from patients with cirrhosis ex vivo here. Treatment with BGB324 did not affect AXL expression (Figs 7A and S10A). Both, BGB324 (1 μM) and metformin (10 mM) treatment restored LPS-induced TNF-α production of monocytes from patients with cirrhosis ex vivo. When comparing AXL⁺ with AXL⁻ monocyte populations from patients with

cirrhosis following metformin treatment, cytokine production was enhanced in AXL⁺ but not AXL⁻ cells (Fig 7B). Viability of monocytes after metformin treatment was marginally reduced (Fig S10B). Whereas phagocytosis capacity of *E. coli* bioparticles was preserved after BGB324 administration, it decreased upon metformin treatment (Fig 7C).

# Discussion

In this work, we detail the characteristics of circulating monocytes in patients suffering from cirrhosis at different stages of disease without signs of AD. We newly describe the expansion of an AXL-expressing immune-regulatory monocyte subset (CD14⁺HLA-DR⁺AXL⁺) along cirrhosis progression and its close association with disease severity, infection susceptibility, development of AD, and prognosis. The AXL-expressing monocyte generation was linked to the abundance of PAMPs and cytokines, phagocytosis, and efferocytosis in the context of recurrent inflammation. Our findings substantially add to the understanding of the pathophysiology of immuneparesis in

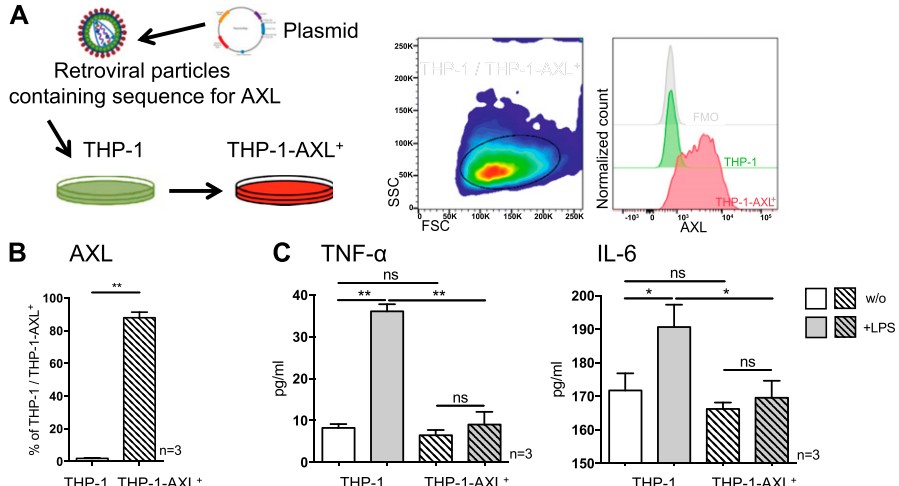

**Figure 5. AXL overexpression in THP-1 cells and LPS-induced inflammatory cytokine production in vitro.**
**(A)** Schematic model of retroviral transduction of THP-1 cells and representative FACS histogram of AXL expression in THP-1-AXL⁺ cells. Side scatter (SSC), forward scatter (FSC). **(B)** AXL expression in THP-1 cells and AXL-expressing THP-1 cells (%). **(C)** TNF-α and IL-6 secretion (pg/ml) in response to LPS in THP-1-AXL⁺ and THP-1 cells. Bar plots showing mean/SD. *P < 0.05/ **P < 0.01 (t tests).

cirrhosis and identify a potential biomarker and immunotherapeutic target.

Although we and others have previously focused on impaired innate immune responses after the onset of AD/ACLF, when infection susceptibility is highest ([10], [12], [18], [20]), it is barely known when and under which circumstances innate immune dysfunction occurs and infection susceptibility emerges during progression of cirrhosis and portal hypertension. Previous studies addressing phenotype and function of classical (CD14⁺CD16⁻) and nonclassical (CD14⁺CD16⁺) monocyte subsets in CLD showed inconsistent data regarding cytokine production and phagocytosis ([31], [32], [33]). We previously showed that inflammatory cytokine production was depressed not only in AD/ACLF but also in stable cirrhosis ([18]), whereas the underlying mechanism remained unexplained.

Here, we demonstrate the accumulation of circulating CD14⁺HLA-DR⁺AXL⁺ monocytes with attenuated innate immune functions, that is, decreased inflammatory cytokine production (TNF-α/IL-6) and T cell activation, along disease progression of cirrhosis in compensated and chronically decompensated patients. Although dysfunctional monocytes were rarely encountered in Child A, their number substantially expanded in advanced stages (Child B/C) and displaced functionally intact monocytes. The findings were irrespective of the underlying aetiology. Monocyte functions were preserved in patients with CLD without cirrhosis (F ≤ 3). Follow-up data of individual patients showed an evolution of AXL-expressing monocytes in parallel with disease severity scores. However, this requires further evaluation in larger prospective longitudinal studies.

AXL is a member of TAM receptors that mainly function as inhibitors of TLR- and cytokine receptor–mediated monocyte/macrophage activation and promoters of apoptotic cell removal ([15], [16]). Loss of AXL expression on antigen-presenting cells has been linked to autoimmunity ([15]). TAM receptors are differentially expressed and exhibit distinct expression, regulation, and activity under specific conditions ([34], [35]). Although CD14⁺HLA-DR⁺AXL⁺ cells accumulated with worsening stages of cirrhosis, they disappeared upon acute hepatic decompensation. By contrast, CD14⁺HLA-DR⁺MERTK⁺ monocytes remained undetectable in stable cirrhosis and emerged upon AD ([18]). In addition, CD14⁺HLA-DR⁺MERTK⁺ cells were abundant in the

circulation and the liver in acute liver failure where they were characterised as resolution-type monocytes/macrophages ([19]). This underlines the distinct and counter-regulatory roles of AXL and MERTK in different phases of cirrhosis and inflammation. Our findings in a human disease verify previous murine data that identified differential and reciprocal expression and function between AXL and MERTK on BMDMs/BMDCs ([34]). The authors described both receptors as phagocytic mediators in vitro, whereas MERTK expression was induced by tolerogenic stimuli and induced tolerance, and AXL was induced by inflammatory stimuli and acted in the feedback inhibition of inflammation ([34]). The underlying differential signalling mechanisms need to be addressed in future investigations.

Extensive characterisation of the AXL-expressing monocyte population revealed an immune-regulatory subset that emerged in the circulation likely to maintain immune homoeostasis despite rising inflammatory signals during progression of cirrhosis. AXL-expressing monocytes were characterised by increased HLA-DR, CD16 and chemokine-receptor expression, enhanced clearance of apoptotic cells, increased expression of Fcγ-receptor, and preserved phagocytosis of E. coli, but attenuated T cell activation and secretion of pro-inflammatory cytokines (TNF-α/IL-6) after microbial challenge presumably in a SOCS1/3-dependent manner, as previously proposed ([16]). Although phagocytosis of pathogens represents the first line of defence when an organism encounters microbes, clearance of apoptotic cells is required for immune homeostasis during inflammation. Our findings suggest that AXL-expressing monocytes may expand during cirrhosis and progressive portal hypertension in response to the uptake of pathogens and bacterial products in the setting of pathologic bacterial translocation ([8], [13], [14], [23]), and to clear apoptotic cell debris accumulating in response to chronic inflammation ([15], [16]). Concurrently, excessive systemic inflammatory responses are inhibited.

Similar to our findings, a recent study described AXL-expressing murine airway macrophages at homeostatic conditions, which increased after influenza infection, thereby preventing excessive tissue inflammation through efferocytosis ([35]). AXL expression was critical for functional compartmentalisation, as it was not present on interstitial lung macrophages ([35]). We propose that in cirrhosis,

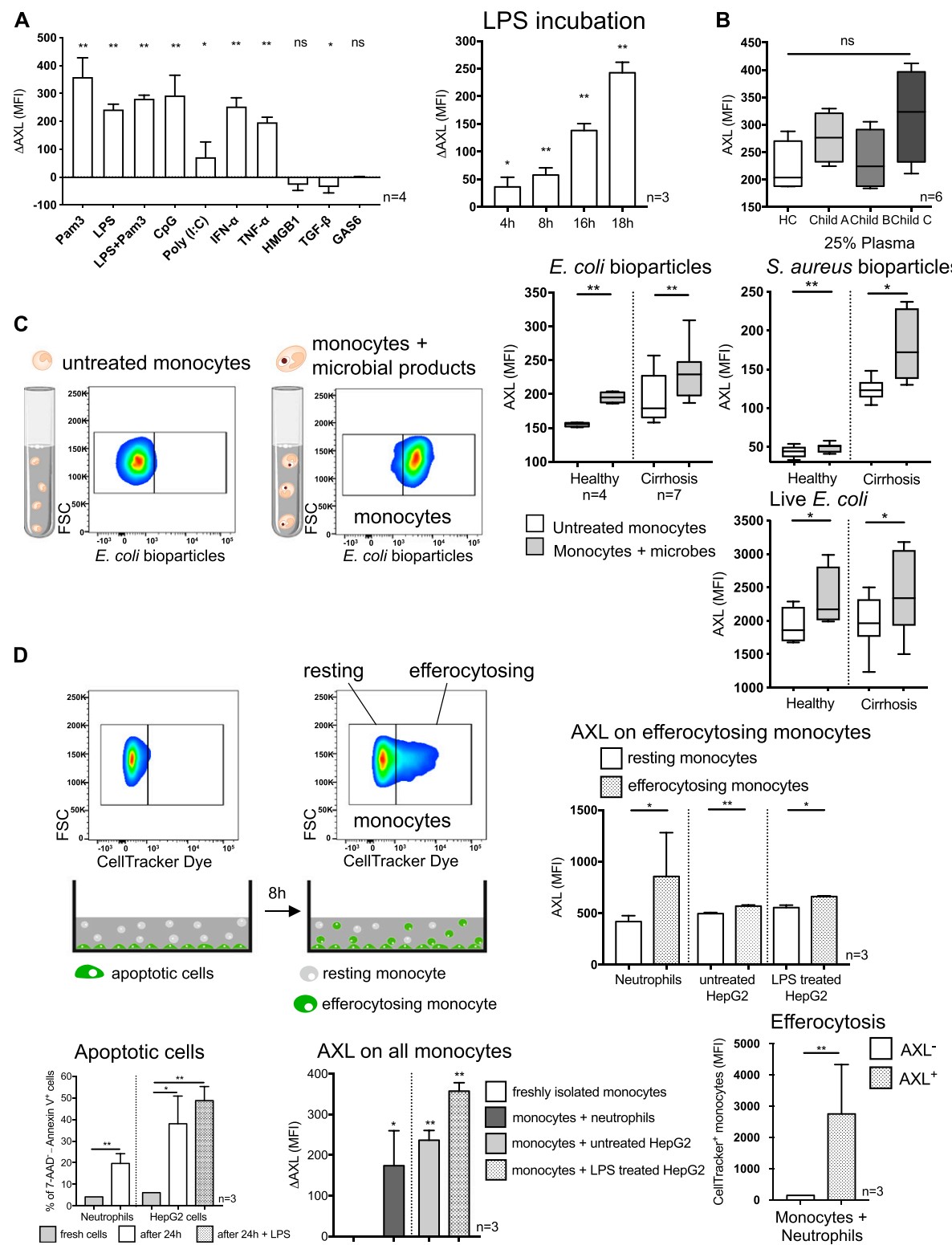

**Figure 6. AXL expression on monocytes in response to bacterial and inflammatory stimuli and following phagocytosis and efferocytosis.**
**(A)** AXL expression after incubation with bacterial/inflammatory stimuli as indicated in vitro for 18 h. Time-dependent effect of LPS on AXL expression. Delta AXL (ΔAXL) MFI shows difference to untreated cells. Bar plots showing mean/SD (*t* tests). **(B)** AXL expression after monocyte incubation in 25% plasma of HCs and patients with cirrhosis for 24 h. **(C)** Representative FACS scatter plots for monocyte phagocytosis of microbial products ex vivo. Forward scatter (FSC). AXL expression after *E. coli* and *S. aureus* bioparticle uptake (15 min) and live GFP-*E. coli* ingestion (60 min) on monocytes from HC and patients with cirrhosis. Box plots showing median/10–90 percentile (Mann–Whitney tests). **(D)** Representative FACS scatter plots for resting (CellTracker⁻) and efferocytosing (CellTracker⁺) monocytes after

AXL may be operative in settings where the nature of injury is driven by excessive pro-inflammatory responses, as present in diverse underlying aetiologies.

This newly described immune-regulatory, AXL-expressing monocyte population must be clearly distinguished from another recently discovered immunosuppressive cell subset that accumulated in the circulation in stages of AD: M-MDSCs were characterised by suppression of T cell activation, pathogen uptake, and TLR-elicited pro-inflammatory responses to microbial challenge (20). Here, we observed that M-MDSCs started to emerge also in patients with cirrhosis without signs of AD. Considering the reduced inflammatory responses to microbial challenge of both CD14$^+$HLA-DR$^+$AXL$^+$- and M-MDSCs compared with functionally intact CD14$^+$HLA-DR$^+$AXL$^-$ cells in relation to their abundance in the circulation, we propose that these populations together largely explain the depressed innate immune responses of the entire monocytic population at these stages.

Having observed AXL up-regulation on circulatory monocytes in relation to portal hypertension, we hypothesized an underlying mechanism involving pathologic bacterial translocation leading to the abundance of bacterial products, PAMPs (8, 13, 14, 23), and subsequent chronic systemic inflammation (8, 23). At the same time, chronic liver injury leads to release of DAMPs (36). Indeed, we were able to generate AXL-expressing monocytes with dampened innate immune responses by stimulation with selected TLR ligands and pro-inflammatory factors in vitro. These findings coincide with previous data showing AXL up-regulation on murine BMDMs (34, 35) and peritoneal macrophages (35) upon stimulation with inflammatory stimuli (34, 35). In contrast to M-MDSCs, generated in vitro by culturing monocytes in ACLF plasma (20), inflammatory factors in plasma of patients with cirrhosis were necessary, but insufficient to induce AXL up-regulation alone. Our data support the hypothesis that efferocytosis and phagocytosis of bacteria in the circulation are further required to enhance AXL up-regulation on monocytes. The stimulatory effect of pathogen uptake on AXL expression is novel and may explain high AXL expression on circulating monocytes in conditions where pathogens and their products become abundant because of pathologic bacterial translocation such as cirrhosis. TAM receptor activation after efferocytosis had previously been shown on murine BMDMs/BMDCs (34).

Dissecting the complexity of differential monocyte differentiation and activation of effector pathways of particular TAM receptors on monocytes at different stages of cirrhosis will be subject to future investigations, including the use of unbiased large-scale techniques. In a multisystem disorder such as cirrhosis, additional compartments such as the liver, but also the gut, the portal circulation, the peritoneum, and potentially others and their tissue-specific immune systems play crucial roles in the pathophysiology of the underlying immuneparesis. It is the aim of our subsequent investigations to detail the differentiation and immune function of tissue-specific myeloid cells in these compartments, in particular in respect to the immune-regulatory role of TAM receptors.

Moreover, by ex vivo proof of principle experiments treating monocytes from cirrhotic patients with the highly specific AXL inhibitor BGB324 and metformin, which was previously described to target and down-regulate AXL (28), innate immune responses were significantly enhanced, suggesting AXL as potential immunotherapeutic target to augment defence against infections. Whereas BGB324 did not negatively affect phagocytic capabilities, metformin did.

BGB324 was originally developed for cancer treatment and is currently tested in clinical Phase Ib/II trials for patients with aggressive and metastatic cancers (37). Interestingly, other studies have examined BGB324 as an anti-fibrotic agent. GAS6/AXL pathways were associated with fibrogenesis in CLD (38) and idiopathic pulmonary fibrosis (39), respectively, and were reversed by BGB324. Multi-tyrosine kinase inhibitors, including AXL, are known for their diverse antitumour effects and are tested in phase III clinical trials for advanced HCC (40). Distinct AXL blockage impacts on tumour progression through immune surveillance by AXL-expressing immune cells and anti-proliferative effect on AXL-expressing tumour cells (37, 40). AXL inhibition by BGB324 may, thus, represent a promising concept with anti-fibrotic, immune-stimulatory, and also anti-tumour effects.

Whereas previous studies described anti-inflammatory, presumably AXL-independent properties of metformin on myeloid cells (41), we observed enhanced immune responses of AXL-expressing monocytes from cirrhotic patients after metformin treatment. Metformin, conventionally used as anti-diabetic drug, exerts various pleiotropic effects acting via diverse downstream signalling pathways (42) and has been reported to be associated with reduced HCC incidence (43) and reduced portal hypertension in cirrhosis models (44). Further studies hint at a potential regulatory effect of metformin on the AXL cascade in the context of cancer (29, 30). As an inexpensive, well-established drug, metformin may represent an interesting immunomodulatory treatment option for patients with cirrhosis and no signs of AD, when AXL-expressing monocytes are frequent and the risk for metformin-associated lactic acidosis is low. Our data are suggestive to further investigate the potential significance of metformin in this context and its underlying signalling mechanism.

As these substances were only tested ex vivo here, subsequent in vivo studies in rodent models are required to systematically investigate target- and off-target effects such as auto-immunity or uncontrolled inflammation. We showed previously that inhibition of MERTK on monocytes of AD/ACLF patients reversed innate immune dysfunction (18). Given the distinct and reciprocal expression profiles of AXL and MERTK in cirrhosis, it further needs to be addressed which receptor to target at which stage of disease and in which compartment.

Finally, strong correlations of AXL expression on monocytes with disease severity and prognosis, that is, i.e. development of infection, episodes of AD, and 1-yr mortality underline its clinical significance. Two recent studies suggested sAXL as a serum biomarker for advanced liver fibrosis, cirrhosis, and HCC (45, 46). Here, we observed strong correlations of AXL-expressing monocytes with

---

co-culture with apoptotic cells for 8 h. Apoptosis of neutrophils and HepG2 cells after 24 h. AXL expression after efferocytosis, AXL expression of resting and efferocytosing monocytes, and efferocytosis capacity for neutrophils of AXL$^+$-/AXL$^-$-monocytes. Bar plots showing mean/SD. *P < 0.05/**P < 0.01 (unpaired/paired t tests).

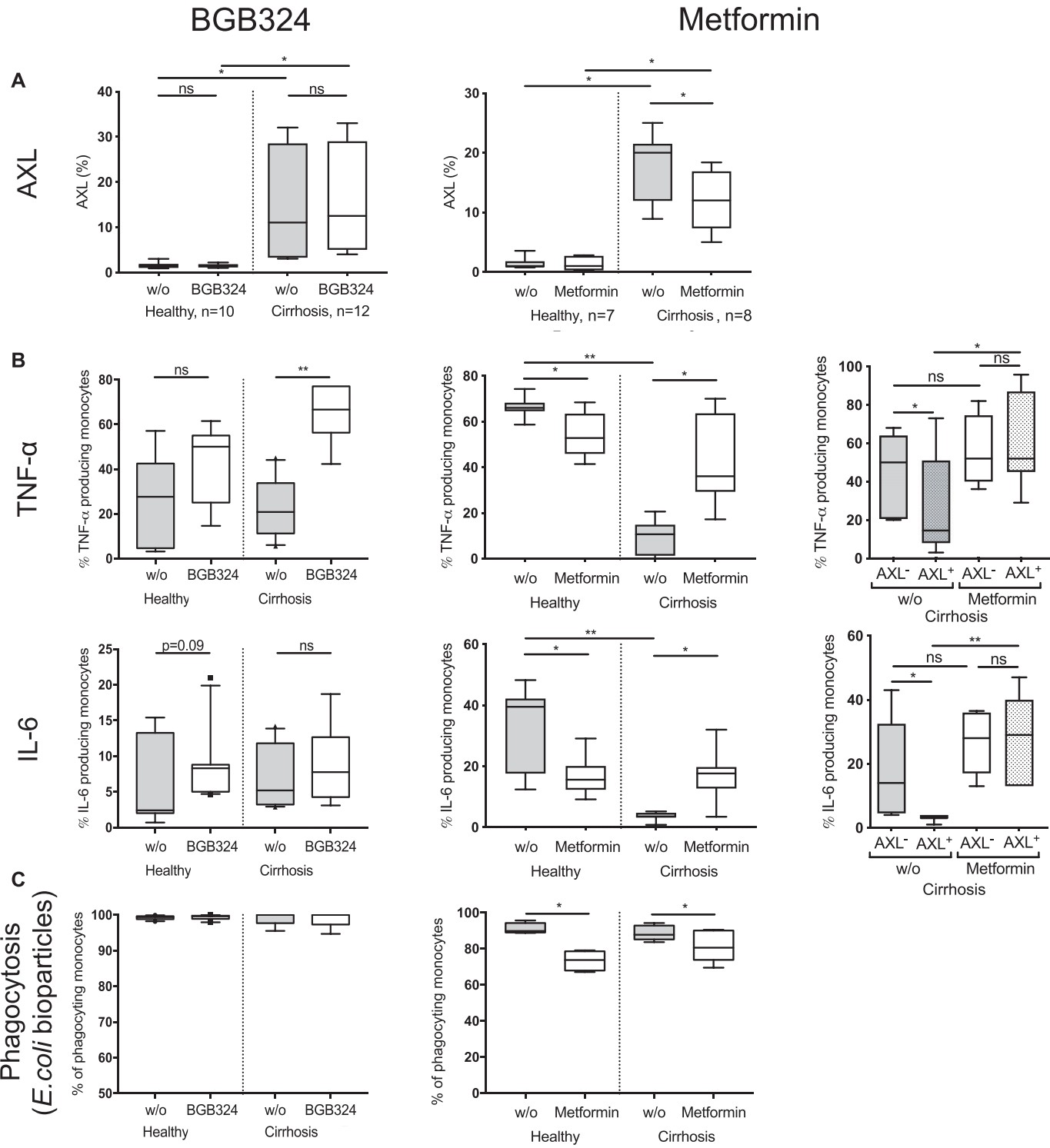

**Figure 7.  Innate immune responses and phagocytosis capacity of monocytes from patients with cirrhosis after AXL inhibition and down-regulation ex vivo.**
**(A, B, C)** AXL expression (% of monocytes), (B) TNF-α and IL-6 production in response to LPS (total monocyte population and AXL⁺/AXL⁻-cells), and (C) monocyte phagocytosis capacity of *E. coli* bioparticles (%CD14⁺ cells) after small molecule inhibitor BGB324 and metformin treatment compared with untreated cells (w/o) in HCs and patients with cirrhosis. Box plots showing median/10–90 percentile. *P < 0.05/**P < 0.01 (Mann–Whitney, Wilcoxon tests).

the shed receptor sAXL and also liver disease severity scores. Based on our findings, the number of AXL-expressing monocytes in blood count may represent a prognostic biomarker for immuneparesis and cirrhosis and validates further evaluation.

In conclusion, the number of AXL-expressing immune-regulatory monocytes in the circulation of patients with cirrhosis indicated disease severity, immuneparesis, infection susceptibility, AD, and mortality. CD14⁺HLA-DR⁺AXL⁺ monocytes were expanded upon PAMP and cytokine exposure, pathogen,- and apoptotic cell uptake and hallmarked by preserved phagocytosis and enhanced efferocytosis but reduced cytokine production and T cell activation, implying a role in immune homeostasis in a condition defined by pathologic bacterial translocation and recurrent inflammation. Immunotherapeutic modulation of AXL may represent an option deserving evaluation to augment immune responses and reduce infection susceptibility, morbidity, and mortality in cirrhosis.

# Materials and Methods

### Patients and sampling

A cohort of 96 patients with cirrhosis was identified at the Cantonal Hospital St. Gallen and the University Hospital Basel, Switzerland, between January 2016 and May 2019. Patients were recruited during consultations (Child-Pugh A [n = 36], B [n = 35], C [n = 17]), respectively, categorised according to Child-Pugh and European Association for the Study of the Liver - Chronic Liver Failure (EASL-CLIF) Consortium scores (47). We included HC (n = 27), patients with CLD without cirrhosis (n = 8; Metavir F ≤ 3), and patients with AD within 24 h following hospital admission (n = 8) as comparators. Healthy volunteers from the regions St. Gallen and Basel, Switzerland, were matched by age and sex and served as HCs. Patients' assent was obtained by the patients' nominated next of kin if they were unable to provide informed consent themselves. Cirrhosis was diagnosed by liver biopsy (n = 92, 95.8%) or clinical presentation with typical ultrasound (n = 4, 4.2%). Exclusion criteria for patients were age younger than 18 yr and evidence of metastatic malignancies (including HCC). Five patients with non-metastatic malignancies were included (HCC, Barcelona Clinic Liver Cancer staging system stages A–B [n = 3]; breast cancer, pT1b, pN0, and M0 [n = 1]; prostate carcinoma, Gleason score 7a [n = 1]). Five patients included with AD had infection at inclusion (spontaneous bacterial peritonitis [n = 3]; spontaneous bacterial peritonitis and urinary tract infection [n = 2]) and six patients included were on immunosuppressive therapy (steroids for AD [n = 3]/adrenal insufficiency [n = 1]/allergy [n = 1]; azathioprine for autoimmune hepatitis [n = 2]). Blood specimens were obtained for ex vivo analysis of monocyte differentiation and function, excessive plasma/serum, and PBMCs were stored. Patients were followed-up for 1 yr for adverse events (infection, development of AD after inclusion, mortality, transplantation, and HCC). Evidence of culture-positive/negative infection was documented. The study had been approved by the local ethics committees (EKSG 15/074/EKNZ 2015-308) and recorded in the clinical trial register ClinicalTrials.gov (identifier: NCT04116242) and Swiss National Clinical Trials Portal (SNCTP000003482).

### Clinical, haematologic, and biochemical parameters

Routine clinical and laboratory parameters obtained by the clinician such as full blood count, CRP, INR, liver, and renal function tests and other variables were entered prospectively into a database. Differential blood count at the sites was performed using Sysmex XE differential analyser (Sysmex Europe GmbH) (Cantonal Hospital St. Gallen) and ADVIA 2120i hematology systems (Siemens Healthineers) (University Hospital Basel).

### Monocyte isolation

Monocytes were isolated from PBMCs using CD14 MicroBeads or Pan Monocyte Isolation Kit (Miltenyi Biotec) as previously described (18). Purity of monocytes was assessed by flow cytometry.

### Flow cytometry–based phenotyping of monocytes, assessment of intracellular cytokine responses to LPS stimulation, and viability assay

Phenotyping of monocytes from blood and isolated PBMCs and measurement of inflammatory cytokine production in response to LPS was undertaken using flow cytometry as previously described (18). Antibodies against CD14, CD16, CD163, CD64, CD11b, chemokine receptor (CCR)5, CCR7 (BD Biosciences), CD32, CX3CR1, TLR2, TLR9 (eBioscience), TLR3 (Invitrogen), HLA-DR, CD15, TLR4, TNF-α, IL-6 (BioLegend), TYRO3, AXL, MERTK, and IFNAR (R&D Systems) were purchased from the indicated companies. In addition to ex vivo phenotyping, TNF-α and IL-6 levels were determined after a 5 h incubation of PBMCs with LPS (100 ng/ml) (Invivogen) in X- VIVO medium without complements (Fig S11; Lonza) in a 37°C, 5% $CO_2$ environment. The Cells were subsequently acquired on BD FACS Canto or BD LSR Fortessa. Flow cytometric gating strategy for circulating monocytes using whole blood or PBMCs was applied as described in reference 48. Flow cytometry data were analysed using FlowJo software (V.10.4.2; Ashland). Results are expressed as the percentage of positive cells and/or MFI.

Cell viability was determined using the Annexin V Apoptosis Detection Kit I (including 7-AAD) according to manufacturer's protocols (BD Biosciences).

### Formula for calculating absolute numbers of TYRO3/AXL/MERTK–expressing cells and M-MDSCs

Absolute cell numbers were calculated with the formula: *Frequency of TYRO3/AXL/MERTK-expressing monocytes or M-MDSCs (% defined by flow cytometry) × monocyte count (G/L)*.

### sAXL, AXL ligands, and cytokines

sAXL (Abcam), GAS6 (Abnova), IL-6, and TNF-α (R&D Systems) were measured using ELISA in plasma or cell culture supernatants as previously described (18).

### Mixed lymphocyte reaction

Monocytes from HCs and patients with cirrhosis expressing high (>20%/450 MFI) versus low (<7.5%/300 MFI) AXL levels were isolated

using the Pan Monocyte Isolation Kit (Miltenyi Biotec) and co-cultured with allogeneic CD3[+] T cells from a different healthy donor, isolated using the Pan T Cell Isolation Kit (Miltenyi Biotec), in a 1:1 ratio. T cell stimulation was induced with anti-CD2/CD3/CD28 beads (T cell Activation/Expansion Kit; Miltenyi Biotec) as previously described (20). T cells were stained with carboxyfluorescein succinimidyl ester at day 0. Proliferation was assessed at day 2 of co-culture by flow cytometry.

### Generation of THP-1 cells stably expressing AXL

pWZL-Neo-Myr-Flag-AXL vector was a kind gift from Hahn's laboratory (#20428; Addgene). Packaging plasmids pUMVC and pMD2.G (a gift from Weinberg's and Trono's laboratories; #8449 and #12259; Addgene) were used for the production of the retrovirus. THP-1 cells were transduced with pWZL-Neo-Myr-Flag-AXL vector, the cells were selected by G418 (Sigma-Aldrich), and THP-1-AXL[+] cells with stably introduced pWZL-Neo-Myr-Flag-AXL were subcloned using MethoCult (StemCell Technologies). The clone with highest AXL expression (for purity see Fig S8) was chosen for phenotypic characterisation by flow cytometry, gene expression analysis by quantitative RT-PCR, and LPS-induced cytokine measurement by ELISA. THP-1 and THP-1-AXL[+] cell lines were cultivated in Roswell Park Memorial Institute 1640 medium (RPMI 1640) (Sigma-Aldrich) supplemented with 10% heat-inactivated FBS, 100 $\mu$g/ml streptomycin, and 100 U/ml penicillin (Sigma-Aldrich).

### Quantitative RT-PCR

Total RNA of isolated monocytes from patients with cirrhosis, HCs, and THP-1/THP-1-AXL[+] cells in cell culture was isolated using RNeasy Mini Kit (QIAGEN) and reversely transcribed into cDNA using High Capacity cDNA Reverse Transcription Kit (Applied Biosystems/Thermo Fisher Scientific). qRT-PCR was performed with 400 ng of cDNA using LightCycler 480 SYBR Green I Master Mix (Roche). Commercial primers for AXL (Hs_AXL_1_SG QuantiTect Primer Assay) were purchased from QIAGEN. Sequences (5′-3′) of primers are as follows: SOCS1 forward: CCC CTT CTG TAG GAT GGT AGC A; reverse: TGC TGT GGA GAC TGC ATT GTC and SOCS3 forward: ATG GTC ACC CAC AGC AAG TT; reverse: TCA CTG CGC TCC AGT AGA AG. GAPDH was used as endogenous control as previously described (35). qRT-PCR was performed according to the manufacturer's recommendations on QuantStudio Real-Time PCR (Applied Biosystems/Thermo Fisher Scientific).

### Whole blood phagocytosis assay

Whole blood was incubated with pHrodo *E. coli* Red BioParticles (Phagocytosis Kit for Flow Cytometry from Invitrogen/Thermo Fisher Scientific) and live *E. coli* NovaBlue carrying the gfp-mut2 encoding plasmid pCD353, which expresses a prokaryotic variant of GFP controlled by a lactac promoter as described in (49) (a kind gift of Prof Dr C Dehio, University of Basel). 5 $\mu$l of *E. coli* Red BioParticles were added for 15 min at 37°C to 100 $\mu$l of whole blood and processed as previously described (20). Blood with bioparticles was stained with antibodies against CD14, CD16, HLA-DR, AXL, and CD15, and CD3, CD19, and CD56 (BD Biosciences) and acquired on the flow

cytometer. *E. coli* bacteria were freshly grown on LB Agar plates supplemented with kanamycin (50 $\mu$g/ml; Sigma-Aldrich) and incubated overnight at 37°C. A single colony was picked and grown in LB medium supplemented with kanamycin (50 $\mu$g/ml) and IPTG (1 mM; Sigma-Aldrich) for GFP induction at 37°C until early logarithmic growth (OD600 = 0.5–0.6) was reached. After the incubation period, bacteria (1 × 10[9] bacteria) were centrifuged at 3000$g$ for 5 min at 4°C, resuspended in 1 ml PBS, and used immediately. 5 × 10[7] of GFP-containing *E. coli* were added for 60 min at 37°C to 100 $\mu$l of whole blood and processed as previously described (20). Blood with GFP-containing *E. coli* was stained with antibodies against CD14, CD16, and AXL and acquired on the flow cytometer. The rate of phagocytosis was obtained by the proportion of GFP positive monocytes.

### In vitro inhibition of AXL

A small-molecule inhibitor of AXL, BGB324 (Selleck Chemicals), and metformin (Stemcell Technologies) were used. Selectivity and mechanism of BGB324 were described previously (27, 38). Metformin was previously described to suppress AXL expression at a concentration of 10 mM (28). We used 0.5 × 10[6] PBMCs from HCs and patients per well on a 48-well plate and cultured them in X-VIVO medium (Lonza) containing 10% FBS in a 37°C, 5% CO$_2$ environment. The cells were treated with 1 $\mu$M BGB324/10 mM metformin or dimethyl sulfoxide/PBS for 24 h, harvested, and washed two times with PBS before the assessment of inflammatory cytokine production in response to LPS (100 ng/ml, 5 h), phagocytosis capacity, and viability of monocytes by flow cytometry. For the assessment of monocyte phagocytosis from isolated PBMCs in vitro, the harvested cells were incubated with pHrodo *E. coli* Red BioParticles (Invitrogen/Thermo Fisher Scientific) for 60 min, processed, and assessed by flow cytometry as previously described (20). The optimal dose of 1 $\mu$M BGB324 was initially defined by a dose finding experiment assessing cytokine production of monocytes in response to LPS (100 ng/ml, 5 h). Cell viability using Annexin V (BD Biosciences) was assessed after BGB324 and metformin treatment (Fig S10B–D).

### In vitro models for the generation of AXL-expressing cells

1 × 10[6] PBMCs per well were cultured on 24-well plates in 1 ml X-VIVO medium (Lonza) in a 37°C, 5% CO$_2$ environment. Cells were stimulated with or without LPS 100 ng/ml (4, 8, 16, 18, 24 h), Pam3CSK4 5 $\mu$g/ml, CpG 10 $\mu$g/ml, poly(I:C) 10 $\mu$g/ml (Invivogen), IFN-$\alpha$ 250 U/ml (Roche), TNF-$\alpha$ 250 U/ml, GAS6 20 nM (R&D Systems), HMGB1 20 ng/ml (Sigma-Aldrich), and TGF-$\beta$ 2 ng/ml (PeproTech) for 18 h. The cells were harvested and subjected to immunophenotyping, intracellular staining of cytokine production in response to LPS and viability assays using flow cytometry as described before.

For the experiments incubating healthy monocytes in plasma from HC (n = 4) and patients with cirrhosis (n = 4), 1 × 10[6] CD14[+] cells were cultured in a 24-well plate for 24 h in X-VIVO medium (Lonza) containing 25% of the indicated plasma in a 37°C, 5% CO$_2$ environment. Subsequently, a fraction of these cells was phenotyped and the remainder was transferred to fresh medium for the assessment of LPS-stimulated TNF-$\alpha$/IL6 production and phagocytosis capacity of monocytes by flow cytometry as detailed before.

For the experiments measuring AXL-expressing monocytes after treatment of bacteria, whole blood was incubated with pHrodo *E. coli* Red BioParticles, pHrodo *S. aureus* Red BioParticles (Invitrogen/Thermo Fisher Scientific) for 15 min, and with live GFP-containing *E. coli* bacteria for 60 min and processed as described above.

### Efferocytosis assay

The experimental design was adapted from Zizzo et al (26) and Triantafyllou et al (19). Human neutrophils were isolated using PolymorphPrep (Axis-Shield) by density-gradient centrifugation according to the manufacturer's protocols, re-suspended at $1 \times 10^6$ cells/ml in RPMI-1640 (Sigma-Aldrich) containing 10% FBS (complete medium), labelled with CellTracker Violet BMQC (5 $\mu$M in serum-free medium, 45 min, 37°C, dark; Life Technologies, Thermo Fisher Scientific), and incubated for 20 h (37°C in 5% $CO_2$) in 300 $\mu$l complete RPMI-1640 in 24-well plates. HepG2 cells were seeded at $0.4 \times 10^6$ cells/ml in 24-well plates, labelled with CellTracker Violet BMQC as described above, and incubated with LPS (1 $\mu$g/ml) for 18 h (37°C in 5% $CO_2$) in 300 $\mu$l complete RPMI-1640. After the incubation period, percentage of apoptotic neutrophils and HepG2 cells in culture was determined using Annexin V Apoptosis Detection Kit I according to the manufacturer's protocols (BD Biosciences). Neutrophils and HepG2 cells were re-suspended in the wells and healthy monocytes were added to apoptotic cells (1:4 monocytes to apoptotic cells ratio) for 8 h in 1 ml fresh complete RPMI-1640 (37°C in 5% $CO_2$). The cells were harvested and washed two times in PBS and subjected to immunophenotyping, intracellular staining of cytokine production in response to LPS (100 ng/ml, 5 h), and viability assays as described above. The rate of efferocytosis was obtained by the proportion of CellTracker-positive monocytes.

### Statistical analyses

Statistical evaluation was performed in GraphPad Prism v.7.0a (GraphPad Software). $P < 0.05$ values were considered statistically significant. Data are shown as box and whiskers or scatter dot plots and expressed as median with 10–90 percentile, unless otherwise specified. For data that did not follow a normal distribution, significance of differences was tested using Mann–Whitney or Wilcoxon tests. Spearman correlation coefficients and area under the receiver operating characteristic curve were calculated. Normally distributed data were compared using paired or unpaired *t*-tests.

## Supplementary Information

## Acknowledgements

The authors are grateful to Prof Dr Jean Pieters and Dr Stefan Wieland for stimulating discussions, Andrej Besse, and Dr Fanny J Lebosse for methodological support; Sylvia Ketterer and Andrijana Bogdanović for their help with patient recruitment; and Prof Dr Christoph Dehio for the kind gift and technical support of GFP-containing *Escherichia coli* bacteria. The authors are also grateful to all patients who consented to take part in this study and all staff at the Cantonal Hospital St. Gallen and University Hospital Basel involved in these patients' care. The authors also thank the Medical Research Center of the Cantonal Hospital St. Gallen and the Department of Biomedicine of the University Hospital Basel for infrastructural support. We finally want to express our gratitude to Dr Harry Antoniades for his important contributions to this project. We will always remember and carry on his enthusiasm and drive for research. The project was supported by the Swiss National Science Foundation (project number 320030_159984) and the Research Committee, Medical Research Centre, Cantonal Hospital St. Gallen (project number 14/17).

## Author Contributions

R Brenig: conceptualization, resources, data curation, formal analysis, investigation, visualization, methodology, and writing—original draft, review, and editing.
OT Pop: resources, formal analysis, investigation, and methodology.
E Triantafyllou: resources, formal analysis, investigation, methodology, and writing—review and editing.
A Geng: resources, formal analysis, investigation, and methodology.
A Singanayagam: formal analysis, investigation, and writing—review and editing.
C Perez-Shibayama: resources, formal analysis, and investigation.
L Besse: resources, formal analysis, investigation, methodology, and writing—review and editing.
J Cupovic: resources, formal analysis, and investigation.
P Künzler: resources and writing—review and editing.
T Boldanova: resources, formal analysis, and investigation.
S Brand: resources, formal analysis, and investigation.
D Semela: resources, formal analysis, and investigation.
FHT Duong: resources, formal analysis, investigation, methodology, and writing—review and editing.
CJ Weston: formal analysis, investigation, methodology, and writing—review and editing.
B Ludewig: resources, formal analysis, and investigation.
MH Heim: resources, formal analysis, investigation, and writing—review and editing.
J Wendon: conceptualization, visualization, and writing—review and editing.
CG Antoniades: conceptualization, formal analysis, investigation, visualization, and methodology.
C Bernsmeier: conceptualization, resources, formal analysis, supervision, funding acquisition, investigation, visualization, methodology, project administration, and writing—original draft, review, and editing.

## Conflict of Interest Statement

The authors declare that they have no conflict of interest.

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
