## [Reviewer comments · Life Science Alliance]

Life Science Alliance

Expression of AXL receptor tyrosine kinase relates to monocyte dysfunction and severity of cirrhosis

Robert Brenig, Oltin Pop, Evangelos Triantafyllou, Anne Geng, Arjuna Singanayagam, Christian Perez-Shibayama, Lenka Besse, Jovana Cupovic, Patrizia Künzler-Heule, Tujana Boldanova, Stephan Brand, David Semela, François Duong, Chris Weston, Burkhard Ludewig, Markus Heim, Julia Wendon, Charalambos Antoniadis, and Christine Bernsmeier

DOI: <https://doi.org/10.26508/lsa.201900465>

Corresponding author(s): Christine Bernsmeier, University Centre for Gastrointestinal and Liver Diseases

Review Timeline:	Submission Date:	2019-06-21
	Editorial Decision:	2019-08-07
	Revision Received:	2019-11-07
	Editorial Decision:	2019-11-27
	Revision Received:	2019-12-01
	Accepted:	2019-12-02

Scientific Editor: Andrea Leibfried

Transaction Report:

August 7, 2019

Re: Life Science Alliance manuscript #LSA-2019-00465-T

Dr. Christine Bernsmeier
University of Basel/ University Hospital Basel, Cantonal Hospital St. Gallen
Basel
Switzerland

Dear Dr. Bernsmeier,

Thank you for submitting your manuscript entitled "Expression of AXL receptor tyrosine kinase relates to monocyte dysfunction and severity of cirrhosis" to Life Science Alliance. The manuscript was assessed by expert reviewers, whose comments are appended to this letter. I apologize for the delay in getting back to you. We were promised a third report on your work repeatedly, but never received it. I therefore decided to move forward with the two reports at hand.

As you will see, the reviewers think that your work remains rather correlative and descriptive and that the number of patients included in the study is on the lower end. However, the reviewers also think that your work may offer interesting avenues for future research.

Given this input, we decided to invite you to submit a revised version of your work to us, addressing the reviewer comments. Importantly, more support is needed for Fig 4 (rev#1) and a control experiment for the poly:I:C treatments needs to get included (rev#3). Furthermore, the phagocytosis assay needs to get replaced by one that is biologically more meaningful.

Thank you for this interesting contribution to Life Science Alliance. We are looking forward to receiving your revised manuscript.

Sincerely,

B. MANUSCRIPT ORGANIZATION AND FORMATTING:

Reviewer #1 (Comments to the Authors (Required)):

Brenig and colleagues explore the association between the expression of the immunoregulatory

TAM receptors -TYRO3, AXL and MERTK - and immunoparesis in monocytes from patients with cirrhosis. The authors have previously identified the expansion of MERTK⁺ monocytes and macrophages in patients with acute on chronic liver failure. Here, they specifically extend their findings to the TAM receptor AXL. The authors detect an expansion of AXL⁺ monocytes in the circulation of patients with cirrhosis that correlates with disease severity. This population was characterized by diminished TNF production upon TLR stimulation as well as decreased ability to activate T cells. In contrast, this population showed enhanced efferocytosis capacity and increased ability to phagocytose E. coli. Interestingly these responses associated with an increased expression of the negative regulators SOCS1 and SOCS3 in AXL⁺ cells. Overall this study further contributes to the characterization of AXL expression in inflammatory diseases, such as in liver disease. While this is an important area of research, the study is mostly correlative. Furthermore, the number of patients for some conditions is limited. However, it is likely that this study could inspire future efforts to determine the potential of AXL expression as a biomarker for survival/disease severity. The following comments are made in order to clarify some of the findings as well as their presentation and discussion in the manuscript.

Figure 1.

Why are the number of monocytes in HC not included in Fig 1A?

Please include a representative flow cytometry staining for TNF α and IL-6.

For figure 1C histogram, please indicate which Child Score is being plotted.

In this figure as well as in multiple others, various panels are neither described in the text or figure legends. Please revise.

Figure 4.

The authors refer in the text that IL-6 production was lower in Axl⁺ monocytes in Figure 4. Can the authors confirm this statement? It is unclear that the % of IL-6⁺ cells is truly reduced in Axl⁺ monocytes (Fig 4A, right panel). Furthermore, for the results in figure 4A, please include not only the % of cytokine positive cells, but also the MFI for the expression of TNF and IL-6.

Figure 6.

Some of the receptors for the PAMPs and DAMPs tested in Figure 6 are either not expressed or expressed at very low levels in human monocytes (ie.: TLR3). As such it would not be expected that such PAMPs/DAMPs would modify the expression of AXL. This should be discussed or the authors should only stimulate human monocytes with the PAMPs/DAMPs they can respond to.

Figure 7.

The authors make use of metformin, which has been shown to reduce Axl expression. However, metformin can have many other pharmacological effects that could regulate cytokine production independently from its ability to modify Axl expression. This should be discussed.

It is unclear why BGB324 did not inhibit phagocytosis of apoptotic cells. Did the concentrations used effectively block AXL kinase activity? Can the authors explain this finding?

It would be interesting to assess if BGB324 and Metformin reduced the amount of expression of SOCS1 and SOCS3.

Other comments:

Please note that AXL does not mediated phagocytosis of bacteria. This should be clarified as the

readers could be confused and associate the increase in both phagocytosis of bacteria and apoptotic cells to an AXL dependent function.

The authors should include further information on the patient characteristics, such as gender.

The authors refer to CXC3R1 as a pro-resolution receptor. However, this is a chemokine receptor. Please revise.

Please, when referring to TYRO3 or PROTEIN S proteins write them in capital letters.

Reviewer #3 (Comments to the Authors (Required)):

The manuscript by Brenig and colleagues describes up-regulated expression of the Axl receptor tyrosine kinase (RTK) in a CD14⁺CD16^{hi}HLA-DR^{hi} circulating monocyte population in human patients with acute-on-chronic liver failure (but without acute decompensation), together with a set of potential functional consequences of this up-regulation for these monocytes. The paper follows upon earlier work by many of these same authors (Bernsmeier et al., *Gastroenterology*, 2015), in which up-regulation of the related RTK Mertk in monocytes from patients with acute decompensation in the context of acute-on-chronic liver failure was also described. The work is carefully presented, but is largely descriptive and correlative. It adds to an ever-expanding literature that addresses Axl up-regulation in the context of liver diseases, including cancer. It raises the possibility that Axl may be a therapeutic target in the context of liver cirrhosis, but does not test this possibility directly.

Specific issues:

1. Background (third paragraph). Axl interaction with the type I interferon receptor has only been demonstrated to be required for immune suppression in dendritic cells. It is not known to be required (and probably is not required) for other TAM functions (e.g., phagocytosis of apoptotic cells) in other TAM-expressing cells (e.g., macrophages). This should be corrected.
2. In general, the authors have examined effects in circulating (blood) monocytes purified from PBMCs. They should discuss the extent to which these cells may be acting in the liver.
3. Many of the effects reported - e.g., Fig. 2A, Fig. 4A - while statistically significant, are small in magnitude and highly variable.
4. The experiments with over-expressing Axl in the THP-1 tumor line (Fig. 5) are confirmatory of a considerable body of previously published work, and do not really add to this literature.
5. Fluorescent bacteria are probably not an appropriate target for the phagocytosis assays (Fig. 4D and elsewhere) reported in the paper. Mertk and Axl have been shown to play important roles in the engulfment of apoptotic cells, a process that requires the exposure of phosphatidylserine on the apoptotic cell surface. How and whether TAM receptors function in the phagocytosis of *E. coli* and other bacteria is not known. In addition, saying that the Axl inhibitor BGB324 has no effect on *E. coli* phagocytosis by monocytes (Fig. 7C) is difficult because the starting value in the assay is 100% - the bacteria may be saturated by monocytes.
6. It is interesting that all of the PAMPs/DAMPs tested display Axl up regulation very similar to what

has already been reported by other investigators in other immune cells - except for poly(I:C), which seems to be incapable of stimulating Axl expression in the monocyte population under study. This is quite curious, since poly(I:C) is among the most potent stimulators of Axl up-regulation in macrophages and DCs. The authors should confirm that the poly(I:C) preparation that they are using is capable of inducing this macrophage/DC up-regulation, because a real difference in Axl regulation between macrophages and monocytes should be followed up.

7. Metformin is a pleiotropic drug with many targets in many pathways. Interpreting its effects on monocytes specifically with respect to Axl is ill-advised.

We would like to thank the reviewers and editorial board for their highly constructive comments that have markedly improved the manuscript.

We have provided a comprehensive revision of our original submission showing that AXL expressing monocytes expand in patients with advanced cirrhosis of the liver and regulate immune responses in relation to disease progression and its prognostic indicators.

As requested, we further detailed in particular the relationship between AXL expression and phagocytosis showing that phagocytosis of apoptotic cells as well as of pathogens were determinants leading to the expansion of an AXL-expressing immune-regulatory monocyte population. AXL expression on monocytes facilitated phagocytosis of apoptotic cells but not of pathogens which echoes previous findings in mice [1–4].

We have addressed each and every comment individually and provided substantial data as outlined below. We hope that this will merit formal acceptance for publication in Life Science Alliance.

Editor's comments:

As you will see, the reviewers think that your work remains rather correlative and descriptive and that the number of patients included in the study is on the lower end. However, the reviewers also think that your work may offer interesting avenues for future research.

Given this input, we decided to invite you to submit a revised version of your work to us, addressing the reviewer comments. Importantly, more support is needed for Fig 4 (rev#1) and a control experiment for the polyI:C treatments needs to get included (rev#3). Furthermore, the phagocytosis assay needs to get replaced by one that is biologically more meaningful.

Response:

Figure 4 was edited to clarify the difference in IL-6 secretion (see response to rev#1).

The experiment using poly(I:C) for the treatment of monocytes in vitro was repeated and amended using monocyte-derived macrophages and dendritic cells as controls (see response to rev#3, Figure 6A, and Figure III below).

The decision to use pHrodo bioparticles to study phagocytosis of pathogens was based on the unique advantage of this test to solely detect internalised particles and not particles binding to surface phagocytosis receptors. Additionally, we added a test using live GFP-*E.coli* bacteria confirming our findings (see response to rev#3, Figure 6C, Supplementary Figure 7C).

Reviewer comments

Reviewer #1

Brenig and colleagues explore the association between the expression of the immunoregulatory TAM receptors -TYRO3, AXL and MERTK - and immunoparesis in monocytes from patients with cirrhosis. The authors have previously identified the expansion of MERTK+ monocytes and macrophages in patients with acute on chronic liver failure. Here, they specifically extend their findings to the TAM receptor AXL. The authors detect an expansion of AXL+ monocytes in the circulation of patients with cirrhosis that correlates with disease severity. This population was characterized by diminished TNF production upon TLR stimulation as well as decreased ability to activate T cells. In contrast, this population showed enhanced efferocytosis capacity and increased ability to phagocytose *E. coli*. Interestingly these responses associated with an increased expression of the negative regulators SOCS1 and SOCS3 in AXL+ cells. Overall this study further contributes to the characterization of AXL expression in inflammatory diseases, such as in liver disease. While this is an important area of research, the study is mostly correlative. Furthermore, the number of patients for some conditions is limited. However, it is likely that this study could inspire future efforts to determine the potential of AXL expression as a biomarker for survival/disease severity. The following comments are made in order to clarify some of the findings as well as their presentation and discussion in the manuscript.

i) Figure 1.

Why are the number of monocytes in HC not included in Fig 1A?

Response:

Numbers of monocytes in healthy subjects define the range of the monocyte count used as a reference to delineate pathological values in clinical practice, for this reason we initially omitted to show the count in this group. We apologise for this inattention. As requested by the reviewer, we now assessed numbers of monocytes also in HC (n=11). Monocyte count of HC was significantly lower compared to patients with liver cirrhosis. Accordingly, figure 1A, supplementary figure 2B and the methods were amended.

Please include a representative flow cytometry staining for TNF α and IL-6.

Response:

Thank you for this comment. As suggested, we additionally included representative flow cytometry staining histograms for TNF- α and IL-6 of HC and patients with cirrhosis stage Child C compared to the corresponding isotype (Figure 1B). Furthermore, in order to systematically report our data in this figure we added LPS-induced TNF- α and IL-6 production of monocytes from patients with acute decompensation of cirrhosis (AD). In patients with AD we had recently described impaired inflammatory cytokine production of circulating monocytes to microbial challenges compared to HC [5,6], which was confirmed in this cohort.

For figure 1C histogram, please indicate which Child Score is being plotted.

Response:

Apologies for not having detailed this point. The representative FACS histogram for AXL expression exemplarily shows a healthy control, a Child C patient and the fluorescence minus one (FMO), as adapted in figure 1C.

In this figure as well as in multiple others, various panels are neither described in the text or figure legends. Please revise.

Response:

Apologies for having described the content of certain figures insufficiently. We systematically reviewed and revised all figures, legends and the corresponding results section in order to precisely detail all the data obtained and graphically depicted.

ii) Figure 4.

The authors refer in the text that IL-6 production was lower in Axl+ monocytes in Figure 4. Can the authors confirm this statement? It is unclear that the % of IL-6+ cells is truly reduced in Axl+ monocytes (Fig 4A, right panel). Furthermore, for the results in figure 4A, please include not only the % of cytokine positive cells, but also the MFI for the expression of TNF and IL-6.

Response:

Thank you for pointing out that the presentation of the data here was misleading. We had initially presented intracellular cytokine production of AXL-expressing monocytes (AXL⁺) versus monocytes not expressing AXL (AXL⁻) on the left panels and on the right panel of distinct monocytic subsets (CD14⁺HLA-DR⁺AXL⁻, CD14⁺HLA-DR⁺AXL⁺ and M-MDSC [which by definition express low HLA-DR; [7]]) from patients with cirrhosis in comparison to CD14⁺HLA-DR⁺AXL⁻ from healthy controls (HC), here representing the majority of monocytes (as shown in Supplementary Figure 6). In order to present the data most clearly and accurately we now restricted the representation to the right panel (see Figure 4A and Supplementary Figure 7A). As requested, we added the MFI for the intracellular production of TNF- α and IL-6 assessed by flow cytometry in addition to the %-age of monocytes (see Supplementary Figure 7). In analogy, we amended the representation of phagocytosis capacity in figure 4D and supplementary figure 7C.

Indeed, in patients with cirrhosis, the IL-6 production of CD14⁺HLA-DR⁺AXL⁺ monocytes was significantly lower, but the difference was numerically modest in %-age (CD14⁺HLA-DR⁺AXL⁺ median 40% (55) vs. CD14⁺HLA-DR⁺AXL⁻ 47% (35.5); median (IQR)) but large in MFI (408 (298) vs. 1790 (681); median (IQR)). The text was amended accordingly.

iii) Figure 6.

Some of the receptors for the PAMPs and DAMPs tested in Figure 6 are either not expressed or expressed at very low levels in human monocytes (ie.: TLR3). As such it would not be expected that such PAMPs/DAMPs would modify the expression of AXL. This should be discussed or the authors should only stimulate human monocytes with the PAMPs/DAMPs they can respond to.

Response:

We thank the reviewer for this important point. It is of course correct that not all of the receptors for these PAMPs/DAMPs tested are expressed equally high on monocytes. The reason why we chose the given selection of PAMPs/DAMPs was on the one hand the pathophysiological consideration that in the context of liver cirrhosis and portal hypertension, pathologic bacterial translocation leads to abundance of PAMPs and subsequent chronic systemic inflammatory responses [8–10]. At the same time chronic liver injury leads to release of DAMPs such as HMBG1 [11].

On the other hand, in the literature the up-regulation of AXL on myeloid cells has previously been reported to be induced by LPS, poly (I:C), IFN- α and IFN- γ [4,12,13]. In supplementary figure 5A, we show the expression of the toll-like receptors for the ligands we chose (TLR2, TLR3, TLR4, TLR9), and interferon- α/β receptor (IFNAR) on monocytes from the patients in this cohort.

TLR3 expression on monocytes although at low levels has been described in the literature before, e.g. [14]. We recently observed lower TLR3 expression in patients with acute decompensation of cirrhosis compared to controls, and TLR3 agonism by poly (I:C) ex vivo modified the differentiation of monocytic cells in this context [6]. As outlined above AXL expression was previously shown to be induced by TLR3 ligand poly(I:C) on murine bone marrow-derived macrophages [4], bone marrow-derived dendritic cells [12], murine peritoneal macrophages, human monocyte-derived macrophages, and airway macrophages [13]. PDGF-receptor (PDGFR- α) is however not expressed on human primary monocytes [15,16], we therefore removed the bar from figure 6A. The discussion of the results has accordingly been reviewed (page 16/17).

iv) Figure 7.

The authors make use of metformin, which has been shown to reduce Axl expression. However, metformin can have many other pharmacological effects that

could regulate cytokine production independently from its ability to modify Axl expression. This should be discussed.

Response:

We appreciate the comment of the reviewer regarding the pleiotropic effects of metformin acting via various downstream signalling pathways [17–19]. This is of course correct, and the data showing AXL down-regulation following metformin treatment and simultaneous enhancement of TNF- α and IL-6 production in monocytes ex vivo from patients with cirrhosis shown here is preliminary and correlative. Thus far, we did not study the signalling cascades underlying our observation.

The idea for this experiment evolved from the publication by Kim et al [20] who showed that metformin reduced the expression of AXL. Further studies hint at a potential regulatory effect of metformin on the AXL cascade in the context of cancer [21,22]. A possible therapeutic use of metformin to restore inflammatory cytokine production in patients with cirrhosis would be highly interesting and our data suggests further investigation of the substance in the context of immune regulation in cirrhosis. The discussion has accordingly been amended (page 18).

An AXL-independent effect of metformin on monocyte cytokine production can not be excluded. Our ex vivo data of monocytes showed that in HC, the percentage of AXL-expressing monocytes was very low and not affected by metformin treatment. TNF- α and IL-6 production of monocytes even tended to be reduced following metformin treatment in HC, as previously described in literature [23]. However, in response to metformin treatment of monocytes from patients with cirrhosis (including 20% of AXL-expressing monocytes), the percentage of AXL-expressing cells was reduced (11%) and cytokine production was enhanced. Interestingly, when comparing AXL⁺- and AXL⁻-monocyte populations of patients with cirrhosis, TNF- α /IL-6 production following metformin treatment was enhanced in AXL⁺- but not AXL⁻-cells. Figure 7A-B, lower panel, and the corresponding result section were amended accordingly. The observed associations hint at a possible AXL-dependent effect of metformin. In supplementary figure 10B, we additionally showed that the viability of monocytes following metformin treatment (10mM) was only marginally reduced.

To underline these data, we also used the THP-1 cell line (harbouring 5% AXL-expressing cells) and an AXL-overexpressing THP-1 cell line (THP-1-AXL⁺, showing

80% AXL-expressing cells) that was transduced by a retrovirus. The percentage of AXL-expressing cells in the two cell lines was reduced by metformin (10mM). In the THP-1 cell line TNF- α production was enhanced following AXL down-regulation by metformin, possibly in an AXL-dependent manner. Metformin treatment was able to reduce but not abolish AXL expression from 80 to 38% of cells in the THP-1-AXL⁺ cell line (see Figure I below).

The underlying signalling mechanism remains to be investigated, using an AXL knockout cell line for instance.

Figure I: AXL expression and pro-inflammatory cytokine production of THP-1 and THP-1-AXL⁺ cells following metformin treatment. AXL expression (%-age and MFI) was measured using flow cytometry. TNF- α and IL-6 production was measured by ELISA (as described in the methods). Delta TNF- α / IL-6 shows the difference between LPS-stimulated and unstimulated cytokine production following metformin (10mM) treatment (24 hours) compared to untreated cells. Mean/SD. t-tests. *p<0.05/**p<0.01.

It is unclear why BGB324 did not inhibit phagocytosis of apoptotic cells. Did the concentrations used effectively block AXL kinase activity? Can the authors explain this finding?

Response:

We apologise for this misconception, but we did not investigate the effect of BGB324 on AXL-dependent efferocytosis of apoptotic cells. According to the manufacturer,

the IC50 of BGB324 is 14nM (<https://www.selleckchem.com/datasheet/r428-S284111-DataSheet.html>). The concentration of 1 μ M has been shown to sufficiently block AXL activity and the downstream target AKT (phospho-AKT) [24,25]. We showed that blocking AXL with 1 μ M BGB324 has the best effect regarding enhancement of TNF- α production. Higher dosage impaired viability of monocytes significantly (see Supplementary Figure 10C-D). In Figure 7, we depict the consequences of the agents on phagocytosis of *E.coli*.

It would be interesting to assess if BGB324 and Metformin reduced the amount of expression of SOCS1 and SOCS3.

Response:

Measuring ex vivo SOCS1 and SOCS3 mRNA expression in monocytes of patients with cirrhosis following BGB324 and metformin treatment is challenging in terms of insufficient yield of mRNA. We therefore measured SOCS1 and SOCS3 expression in healthy monocytes upon BGB324 (1 μ M)/ metformin (10mM) incubation (24 hours). SOCS3 expression was significantly lower following BGB324 treatment and elevated after metformin treatment compared to untreated monocytes. SOCS1 expression was unchanged (see Figure II below). The underlying signalling mechanism of BGB324 and metformin action in monocytes remain insufficiently understood. Whether they are SOCS1- and SOCS3-dependent remains to be investigated.

Figure II: SOCS1 and SOCS1 mRNA expression of healthy monocytes following BGB324 and metformin treatment. SOCS1/3 expression was measured using quantitative RT PCR (as described in the methods) following metformin (10mM) and BGB324 (1 μ M) treatment (24 hours) compared to untreated cells. Mean/SD. T-tests. *p<0.05/**p<0.01.

Other comments:

Please note that AXL does not mediated phagocytosis of bacteria. This should be clarified as the readers could be confused and associate the increase in both phagocytosis of bacteria and apoptotic cells to an AXL dependent function.

Response:

We appreciate this comment and realise that we did not fully describe our experimental approach. This point will be detailed in response to comment 5 of reviewer #3 (see below). We systematically reviewed all figures, legends and the corresponding passages in the results and discussion sections and have made amendments to clarify our interpretation of the results.

The authors should include further information on the patient characteristics, such as gender.

Response:

As suggested by the reviewer, gender characteristics have been appended to supplementary table 1 and the corresponding passages in the result sections. Additionally, we added the underlying aetiologies of cirrhosis to supplementary table 2.

The authors refer to CXC3R1 as a pro-resolution receptor. However, this is a chemokine receptor. Please revise.

Response:

The particular sections with this misleading description have been revised.

Please, when referring to TYRO3 or PROTEIN S proteins write them in capital letters.

Response:

The correct nomenclature of TYRO3 and PROTEIN S has been revised in all figures and text passages.

Reviewer #3:

The manuscript by Brenig and colleagues describes up-regulated expression of the Axl receptor tyrosine kinase (RTK) in a CD14⁺CD16^{hi}HLA-DR^{hi} circulating monocyte population in human patients with acute-on-chronic liver failure (but without acute decompensation), together with a set of potential functional consequences of this up-regulation for these monocytes. The paper follows upon earlier work by many of these same authors (Bernsmeier et al., *Gastroenterology*, 2015), in which up-regulation of the related RTK Merck in monocytes from patients with acute decompensation in the context of acute-on-chronic liver failure was also described. The work is carefully presented, but is largely descriptive and correlative. It adds to an ever-expanding literature that addresses Axl up-regulation in the context of liver diseases, including cancer. It raises the possibility that Axl may be a therapeutic target in the context of liver cirrhosis, but does not test this possibility directly.

Specific issues:

1. *Background (third paragraph). Axl interaction with the type I interferon receptor has only been demonstrated to be required for immune suppression in dendritic cells. It is not known to be required (and probably is not required) for other TAM functions (e.g., phagocytosis of apoptotic cells) in other TAM-expressing cells (e.g., macrophages). This should be corrected.*

Response:

We thank the reviewer for the critical review of the cited literature and corrected the paragraph in the background section (page 6).

2. *In general, the authors have examined effects in circulating (blood) monocytes purified from PBMCs. They should discuss the extent to which these cells may be acting in the liver.*

Response:

We thank reviewer #3 for pointing out the importance of other compartments involved in the pathogenesis of immunoparesis in cirrhosis. Patients with cirrhosis develop defective host defence mechanisms leading to infectious complications, which are

known to accelerate disease progression and lead to decompensation and death. For this reason, we primarily focused on circulating monocytes, which we considered essential for the primary host defence against bacterial infection of diverse tissues and/or sepsis. For the development of immunoparesis many other compartments and their tissue-specific immune systems play crucial roles such as the liver, but also the gut, the portal circulation, the peritoneum and potentially others.

In the liver, we know that MERTK-expressing macrophages increase in advanced stages of cirrhosis, especially following decompensation with organ failure (ACLF; [5]). Given the reciprocal regulation of the TAM receptors MERTK and AXL in other conditions [4,13,26], we speculate that AXL may be expressed on monocytes and macrophages and regulate tissue homeostasis in stages, where MERTK is not abundant.

It is the aim of our subsequent investigations to detail the immune function of tissue-specific macrophages in the liver and other compartments, and in particular to delineate the role of TAM receptors there. We added a section in the discussion (page 17) detailing our planned future investigation of macrophages from the liver and diverse compartments involved in the pathogenesis of immunoparesis in cirrhosis as a systemic condition.

3. Many of the effects reported - e.g., Fig. 2A, Fig. 4A - while statistically significant, are small in magnitude and highly variable.

Response:

We agree that some of the differences observed are numerically moderate, which however does not exclude a biologically relevant effect.

In figure 2A we observe a gradual but significant increase of AXL expressing monocytes over the distinct stages of cirrhosis severity.

In figure 4A, in patients with cirrhosis, we observe declines of TNF- α -producing monocytes from 57% (47.3) to 40% (34.6) (AXL⁻ vs. AXL⁺; median (IQR)) and of IL-6-producing monocytes (see rev#1, question ii)), which is a dramatic change in monocyte biology. The data was substantiated by adding the MFI (see also response to rev#1, question ii)). Also, relating the numerically small reduction of inflammatory cytokine production in response to LPS of both CD14⁺HLA-DR⁺AXL⁺- and M-MDSCs compared to CD14⁺HLA-DR⁺AXL⁻ cells to their abundance in the circulation (Figure

1C, Supplementary Figure 6A-B), these two subpopulations together probably explain a major proportion of the depressed innate immune responses observed in the entire monocytic population in cirrhosis (Figure 1B).

The variability of the data is due to the heterogenous nature and aetiologies underlying liver cirrhosis. We did not restrict our study to certain aetiologies as we aim to understand common mechanisms of this disease condition.

4. The experiments with over-expressing Axl in the THP-1 tumor line (Fig. 5) are confirmatory of a considerable body of previously published work, and do not really add to this literature.

Response:

Thank you for critically scrutinising our in vitro data. We agree that using this newly established AXL-overexpressing THP-1 cell line model (Figure 5) does not substantially contribute novel information to the immunosuppressive effects of the AXL receptor. The model reproduced the impaired inflammatory cytokine production observed in primary human AXL-expressing monocytes from patients with cirrhosis ex vivo as we would have expected from the literature. Given the unphysiologically high AXL expression of THP-1-AXL⁺ cells under retroviral transduction, which largely exceeds the AXL expression observed in patients with cirrhosis ex vivo, this model may not represent the pathophysiological in vivo situation to the full extent. However, the model might be useful to investigate mechanisms in vitro in the future.

5. Fluorescent bacteria are probably not an appropriate target for the phagocytosis assays (Fig. 4D and elsewhere) reported in the paper. Mertk and Axl have been shown to play important roles in the engulfment of apoptotic cells, a process that requires the exposure of phosphatidylserine on the apoptotic cell surface. How and whether TAM receptors function in the phagocytosis of E. coli and other bacteria is not known. In addition, saying that the Axl inhibitor BGB324 has no effect on E. coli phagocytosis by monocytes (Fig. 7C) is difficult because the starting value in the assay is 100% - the bacteria may be saturated by monocytes.

Response:

Thank you for raising this important issue. We carefully reflected our data reporting phagocytosis capacity of monocytes and our conclusions here.

As you highlighted, the dogma is that AXL facilitates efferocytosis, i.e. phagocytosis of apoptotic cells, but is dispensable for phagocytosis of pathogens as shown by the group of Williams et al.[3]. In this paper, phagocytosis capacity was measured using a protocol involving live GFP-containing *E.coli* uptake in murine bone marrow-derived and peritoneal macrophages as well as fluorescent *E.coli*- and *S.aureus*-bioparticles. Here, we investigated the function of monocytes expressing AXL, which expanded in cirrhosis and are barely present in the circulation of healthy humans. We reported the distinct differentiation of these monocytes and their function. It became obvious that although the population showed reduced cytokine production and T cell activation, phagocytosis capacities (measured by fluorescent pHrodo *E.coli* bioparticles® [Invitrogen/Thermo Fisher Scientific]) were preserved (in %-age and MFI, Figure 4 and Supplementary Figure 7). We initially over-interpreted our data concluding “enhanced phagocytosis capacity” of this distinct population which is not quite accurate. What we reliably observed is an increase in AXL expression following *E.coli* bioparticle uptake rather than enhanced phagocytosis capacity (Figures 4D and 6C, Supplementary Figure 7C/D). We additionally performed experiments using pHrodo *S. aureus* bioparticles which showed the same results (see Figure 6C).

In order to confirm our data by a different test we used live GFP-*E.coli* as described in the paper by Williams et al.[3] (see also method section). The experiment indeed confirms our findings obtained with *E.coli* bioparticles: Phagocytosis capacity was preserved in cirrhosis patients compared to HC; and phagocytosis led to an up-regulation of AXL expression (Figure 6C, Supplementary Figure 7C). The experiment actually does not differentiate between GFP-*E.coli* bound to surface receptors and engulfed GFP-*E.coli* (in contrast to the pHrodo bioparticle experimental approach which only detects internalized particles). These data suggest that besides the effect of PAMPs on AXL up-regulation, there may thus be an effect of bacterial uptake inducing AXL up-regulation on monocytes.

The relevant paragraphs in the methods, results and discussion sections were revised accordingly and substantially improved the manuscript.

6. It is interesting that all of the PAMPs/DAMPs tested display Axl up regulation very similar to what has already been reported by other investigators in other immune

cells - except for poly(I:C), which seems to be incapable of stimulating Axl expression in the monocyte population under study. This is quite curious, since poly(I:C) is among the most potent stimulators of Axl up-regulation in macrophages and DCs. The authors should confirm that the poly(I:C) preparation that they are using is capable of inducing this macrophage/DC up-regulation, because a real difference in Axl regulation between macrophages and monocytes should be followed up.

Response:

Thank you for indicating this discrepancy between recent literature [4,12,13] and our reported results. We repeated the experiments using poly(I:C) on human monocytes in comparison to human dendritic cells and human monocyte-derived macrophages. We acknowledge that our initially reported results were incomplete. Poly(I:C) strongly induced AXL expression on monocyte-derived macrophages and significantly but to a considerably lower extent also on monocytes and dendritic cells (see Figure III below) in accordance with recent data [4,12]. Figure 6A was amended accordingly.

Figure III: Poly(I:C) induces AXL up-regulation on human monocytes, monocyte-derived macrophages, and dendritic cells. Difference of AXL (delta AXL [MFI]) expression between poly(I:C)-treated and untreated cells is shown. PBMCs and macrophages were stimulated with poly(I:C), 10µg/mL, for 18 hours and compared to untreated cells (as described in methods). AXL expression on DCs was determined using following receptors (AXL, CD14, CD16, HLA-DR, CD123, CD11c, CD1c, CD141, and lineage markers [CD3, CD19, CD20, CD56]). M0 Macrophages were derived from human monocytes using following protocol: To generate M0 macrophages, magnetic bead-enriched CD14+ monocytes were treated for 5 days with recombinant human macrophage colony-stimulating factor (M-CSF; 25 ng/mL; Peprotech) in RPMI 1640, 10% fetal bovine serum, 1% penicillin-streptomycin. To

confirm macrophage differentiation, cell morphology using conventional microscopy, MERTK-, and CD11b expression was assessed using flow cytometry. Mean/SD. T-tests. *p<0.05/**p<0.01.

7. Metformin is a pleiotropic drug with many targets in many pathways. Interpreting its effects on monocytes specifically with respect to Axl is ill-advised.

Response:

We thank both reviewers for this comment and strongly agree. This point has been addressed in response to iv), reviewer #1 above.

References

- [1] Scott RS, McMahon EJ, Pop SM, Reap EA, Caricchio R, Cohen PL, Earp HS, Matsushima GK (2001) Phagocytosis and clearance of apoptotic cells is mediated by MER. *Nature* 411: 207–211.
- [2] Lemke G, Rothlin CV (2008) Immunobiology of the TAM receptors. *Nat Rev Immunol* 8: 327–336.
- [3] Williams JC, Craven RR, Earp HS, Kawula TH, Matsushima GK (2009) TAM receptors are dispensable in the phagocytosis and killing of bacteria. *Cell Immunol* 259: 128–134.
- [4] Zagórska A, Través PG, Lew ED, Dransfield I, Lemke G (2014) Diversification of TAM receptor tyrosine kinase function. *Nat Immunol* 15: 920–928.
- [5] Bernsmeier C, Pop OT, Singanayagam A, Triantafyllou E, Patel VC, Weston CJ, Curbishley S, Sadiq F, Vergis N, Khamri W, et al. (2015) Patients with acute-on-chronic liver failure have increased numbers of regulatory immune cells expressing the receptor tyrosine kinase MERTK. *Gastroenterology* 148: 603-615.e14.
- [6] Bernsmeier C, Triantafyllou E, Brenig R, Lebosse FJ, Singanayagam A, Patel VC, Pop OT, Khamri W, Nathwani R, Tidswell R, et al. (2018) CD14+ CD15– HLA-DR– myeloid-derived suppressor cells impair antimicrobial responses in patients with acute-on-chronic liver failure. *Gut* 67: 1155–1167.
- [7] Bronte V, Brandau S, Chen S-H, Colombo MP, Frey AB, Greten TF, Mandruzzato S, Murray PJ, Ochoa A, Ostrand-Rosenberg S, et al. (2016) Recommendations for myeloid-derived suppressor cell nomenclature and characterization standards. *Nat Commun* 7: 12150.
- [8] Hartmann P, Seebauer CT, Schnabl B (2015) Alcoholic liver disease: The gut microbiome and liver crosstalk. *Alcohol Clin Exp Res* 39: 763–775.
- [9] Seo YS, Shah VH (2012) The role of gut-liver axis in the pathogenesis of liver cirrhosis and portal hypertension. *Clin Mol Hepatol* 18: 337–346.
- [10] Steib CJ, Schewe J, Gerbes AL (2015) Infection as a Trigger for Portal Hypertension. *DDI* 33: 570–576.
- [11] Hernandez C, Huebener P, Pradere J-P, Friedman RA, Schwabe RF (2019) HMGB1 links chronic liver injury to progenitor responses and hepatocarcinogenesis. *J Clin Invest* 128: 2436–2450.
- [12] Rothlin CV, Ghosh S, Zuniga EI, Oldstone MBA, Lemke G (2007) TAM

- receptors are pleiotropic inhibitors of the innate immune response. *Cell* 131: 1124–1136.
- [13] Fujimori T, Grabiec AM, Kaur M, Bell TJ, Fujino N, Cook PC, Svedberg FR, MacDonald AS, Maciewicz RA, Singh D, et al. (2015) The Axl receptor tyrosine kinase is a discriminator of macrophage function in the inflamed lung. *Mucosal Immunol* 8: 1021–1030.
- [14] Monguió-Tortajada M, Franquesa M, Sarrias M-R, Borràs FE (2018) Low doses of LPS exacerbate the inflammatory response and trigger death on TLR3-primed human monocytes. *Cell Death Dis* 9: 1–14.
- [15] Inaba K, Suzuki M, Maegawa K, Akiyama S, Ito K, Akiyama Y (2008) A pair of circularly permuted PDZ domains control RseP, the S2P family intramembrane protease of *Escherichia coli*. *J Biol Chem* 283: 35042–35052.
- [16] Chan LF, Webb TR, Chung T-T, Meimaridou E, Cooray SN, Guasti L, Chapple JP, Egertová M, Elphick MR, Cheetham ME, et al. (2009) MRAP and MRAP2 are bidirectional regulators of the melanocortin receptor family. *PNAS* 106: 6146–6151.
- [17] Rena G, Hardie DG, Pearson ER (2017) The mechanisms of action of metformin. *Diabetologia* 60: 1577–1585.
- [18] Hattori Y, Hattori K, Hayashi T (2015) Pleiotropic Benefits of Metformin: Macrophage Targeting Its Anti-inflammatory Mechanisms. *Diabetes* 64: 1907–1909.
- [19] Ursini F, Russo E, Pellino G, D'Angelo S, Chiaravalloti A, De Sarro G, Manfredini R, De Giorgio R (2018) Metformin and Autoimmunity: A “New Deal” of an Old Drug. *Front Immunol* 9:.
- [20] Kim N-Y, Lee H-Y, Lee C (2015) Metformin targets Axl and Tyro3 receptor tyrosine kinases to inhibit cell proliferation and overcome chemoresistance in ovarian cancer cells. *Int J Oncol* 47: 353–360.
- [21] Bansal N, Petrie K, Christova R, Chung C-Y, Leibovitch BA, Howell L, Gil V, Sbirkov Y, Lee E, Wexler J, et al. (2015) Targeting the SIN3A-PF1 interaction inhibits epithelial to mesenchymal transition and maintenance of a stem cell phenotype in triple negative breast cancer. *Oncotarget* 6: 34087–34105.
- [22] Fujimori T, Kato K, Fujihara S, Iwama H, Yamashita T, Kobayashi K, Kamada H, Morishita A, Kobara H, Mori H, et al. (2015) Antitumor effect of metformin on cholangiocarcinoma: In vitro and in vivo studies. *Oncol Rep* 34: 2987–

- 2996.
- [23] Kim J, Kwak HJ, Cha J-Y, Jeong Y-S, Rhee SD, Kim KR, Cheon HG (2014) Metformin suppresses lipopolysaccharide (LPS)-induced inflammatory response in murine macrophages via activating transcription factor-3 (ATF-3) induction. *J Biol Chem* 289: 23246–23255.
 - [24] Bárcena C, Stefanovic M, Tutusaus A, Joannas L, Menéndez A, García-Ruiz C, Sancho-Bru P, Marí M, Caballeria J, Rothlin CV, et al. (2015) Gas6/Axl pathway is activated in chronic liver disease and its targeting reduces fibrosis via hepatic stellate cell inactivation. *J Hepatol* 63: 670–678.
 - [25] Holland SJ, Pan A, Franci C, Hu Y, Chang B, Li W, Duan M, Torneros A, Yu J, Heckrodt TJ, et al. (2010) R428, a selective small molecule inhibitor of Axl kinase, blocks tumor spread and prolongs survival in models of metastatic breast cancer. *Cancer Res* 70: 1544–1554.
 - [26] Seitz HM, Camenisch TD, Lemke G, Earp HS, Matsushima GK (2007) Macrophages and dendritic cells use different Axl/Mertk/Tyro3 receptors in clearance of apoptotic cells. *J Immunol* 178: 5635–5642.

November 27, 2019

RE: Life Science Alliance Manuscript #LSA-2019-00465-TR

Dr. Christine Bernsmeier
University of Basel/ University Hospital Basel, Cantonal Hospital St. Gallen
Basel
Switzerland

Dear Dr. Bernsmeier,

Thank you for submitting your revised manuscript entitled "Expression of AXL receptor tyrosine kinase relates to monocyte dysfunction and severity of cirrhosis". As you will see, reviewer #1 appreciates the introduced changes and we would thus be happy to publish your paper in Life Science Alliance pending final revisions necessary to meet our formatting guidelines:

- Please link your profile in our submission system to your ORCID iD, you should have received an email with instructions on how to do so
- Please add labels to all FACS scatter plots shown. The ones shown in figure 1A and 3A are partially the same - please clarify / amend.

A. FINAL FILES:

-- Summary blurb (enter in submission system): A short text summarizing in a single sentence the study (max. 200 characters including spaces). This text is used in conjunction with the titles of papers, hence should be informative and complementary to the title. It should describe the context and significance of the findings for a general readership; it should be written in the present tense

and refer to the work in the third person. Author names should not be mentioned.

B. MANUSCRIPT ORGANIZATION AND FORMATTING:

Sincerely,

Reviewer #1 (Comments to the Authors (Required)):

The authors have satisfactorily addressed the comments raised during the revision process. I have no further comments.

Response to editor

Dear Dr. Leibfried,

We would like to thank the reviewers and the editorial board very much for accepting our manuscript for publication in "Life Science Alliance".

Please find answers to your requests below:

- Please link your profile in our submission system to your ORCID iD, you should have received an email with instructions on how to do so
- As requested, I linked my profile to my ORCID iD.

- Please add labels to all FACS scatter plots shown. The ones shown in figure 1A and 3A are partially the same - please clarify / amend.
- We added labels to all FACS scatter plots and histograms. This applies to Figures 1,3,5,6 and Supplementary Figures 2 and 7. The duplicate scatter plot in Figure 3A (first plot top left) has been removed.

December 2, 2019

RE: Life Science Alliance Manuscript #LSA-2019-00465-TRR

Dr. Christine Bernsmeier
University Centre for Gastrointestinal and Liver Diseases
Petersgraben 4
Basel 4031
Switzerland

Dear Dr. Bernsmeier,

Thank you for submitting your Research Article entitled "Expression of AXL receptor tyrosine kinase relates to monocyte dysfunction and severity of cirrhosis". It is a pleasure to let you know that your manuscript is now accepted for publication in Life Science Alliance. Congratulations on this interesting work.

*****IMPORTANT:** If you will be unreachable at any time, please provide us with the email address of an alternate author. Failure to respond to routine queries may lead to unavoidable delays in publication.*******

DISTRIBUTION OF MATERIALS:

Again, congratulations on a very nice paper. I hope you found the review process to be constructive and are pleased with how the manuscript was handled editorially. We look forward to future exciting submissions from your lab.

Sincerely,
